# Genetic variation in the immunoglobulin heavy chain locus shapes the human antibody repertoire

Oscar L. Rodriguez[1], Yana Safonova [2], Catherine A. Silver[1], Kaitlyn Shields[1], William S. Gibson[1], Justin T. Kos [1], David Tieri[1], Hanzhong Ke [3,4], Katherine J. L. Jackson [5], Scott D. Boyd [6], Melissa L. Smith [1]✉, Wayne A. Marasco [3,4]✉ & Corey T. Watson [1]✉

Variation in the antibody response has been linked to differential outcomes in disease, and suboptimal vaccine and therapeutic responsiveness, the determinants of which have not been fully elucidated. Countering models that presume antibodies are generated largely by stochastic processes, we demonstrate that polymorphisms within the immunoglobulin heavy chain locus (IGH) impact the naive and antigen-experienced antibody repertoire, indicating that genetics predisposes individuals to mount qualitatively and quantitatively different antibody responses. We pair recently developed long-read genomic sequencing methods with antibody repertoire profiling to comprehensively resolve IGH genetic variation, including novel structural variants, single nucleotide variants, and genes and alleles. We show that IGH germline variants determine the presence and frequency of antibody genes in the expressed repertoire, including those enriched in functional elements linked to V(D)J recombination, and overlapping disease-associated variants. These results illuminate the power of leveraging IGH genetics to better understand the regulation, function, and dynamics of the antibody response in disease.

Antibodies (Abs) are critical to the function of the adaptive immune system and have evolved to be one of the most diverse protein families in the human body, providing essential protection against foreign pathogens. The circulating Ab repertoire is composed of hundreds of millions of unique Abs[1,2], and the composition of the repertoire varies considerably between individuals[1–3], potentially explaining the varied Ab responses observed in a variety of disease contexts, including infection[4–8], autoimmunity[9–12], and cancer[13–15]. The initial formation of the Ab repertoire is mediated by complex molecular processes, and can be influenced by factors such as prior vaccination and infection, health status, sex, age, and genetics[16–21]. Delineating the mechanisms

that drive variation in the functional Ab response is critical not only to understanding B cell-mediated immunity in disease, but also ultimately informing the design of improved vaccines and therapies[22,23]. With respect to genetic factors, the impact of variants in the immunoglobulin heavy (IGH) and light chain loci on the antibody response has not been determined.

The human IGH locus is located immediately adjacent to the telomere of chromosome 14, and harbors 129 variable (V), 27 diversity (D) and 9 joining (J) genes that are utilized during V(D)J recombination to produce the heavy chain of an Ab[24]. The IGH locus is now understood to be among the most polymorphic and complex regions of the human

[1]Department of Biochemistry and Molecular Genetics, University of Louisville School of Medicine, Louisville, KY, USA. [2]Department of Computer Science, Johns Hopkins University, Baltimore, MD, USA. [3]Department of Cancer Immunology and Virology, Dana-Farber Cancer Institute, Harvard Medical School, Boston, MA, USA. [4]Department of Medicine, Harvard Medical School, Boston, MA, USA. [5]The Garvan Institute of Medical Research, Darlinghurst, NSW, Australia. [6]Department of Pathology, Stanford University School of Medicine, Stanford, CA, USA. ✉e-mail: ml.smith@louisville.edu; Wayne_Marasco@dfci.harvard.edu; corey.watson@louisville.edu

genome[3,25–29]. Akin to the extensive genetic diversity observed in the human leukocyte antigen (HLA) locus, >680 IGH alleles have been cataloged solely from limited surveys[30]. In addition, IGH is highly enriched for large structural variants (SVs), including insertions, deletions, and duplications of functional genes, many of which show considerable variability between human populations[25,29]. This extensive haplotype diversity and locus structural complexity has made IGH haplotype characterization challenging using standard high-throughput approaches, and as a result it has been largely ignored by genome-wide studies[25,28,31]. This has hindered our ability to assess the contribution of IGH polymorphism in disease phenotypes, and more fundamentally, our ability to conduct functional/molecular studies. We currently understand little about the genetic factors, and thus the associated molecular mechanisms, that dictate the regulation of the human Ab response. In fact, much of what we understand about the specific genomic factors involved in Ab repertoire development and variability comes from inbred animal models[32–35], even though such questions would have greater relevance to health if addressed in outbred human populations[22]. These limitations continue to impede our understanding of the contribution of IGH polymorphism to disease risk, infection and response to vaccines and therapeutics[22,31,36,37].

Several lines of evidence now support the importance of IGH genetic variation in human B cell-mediated immune responses. Studies in monozygotic (MZ) twins have shown that many Ab repertoire features are correlated within twin pairs in both naïve and antigen-experienced B cell subsets, indicating strong heritable factors underlying repertoire variability[20,21,38]. Other studies have demonstrated that specific SVs and IG coding and regulatory element polymorphisms contribute to inter-individual variability in expressed human Ab repertoires[23,39–42]. These observations, alongside biases in IG gene usage in various disease contexts, underscore potential connections between the germline and Ab function[39,41,43,44]. Importantly, in many cases, key functional amino acids identified in disease-associated and antigen-specific Abs are encoded by polymorphic positions with variable allele frequencies among populations[23,41]. These observations indicate that IGH variants could offer direct translational opportunities, with the ability to subset the population according to IG genotypes for more tailored healthcare decisions[22]. However, investigations of the direct functional effects of human IGH germline variation conducted to date have been limited to only a small fraction of IGH variants known[39–42].

Here, to identify IGH polymorphisms that affect variation in the expressed Ab repertoire, we perform long-read sequencing to comprehensively genotype IGH and combine these data with adaptive immune receptor repertoire sequencing (AIRR-seq) in 154 healthy individuals. From these data, we detect an extensive number of single nucleotide variants (SNVs), small insertion-deletions (indels) and SVs across IGH, including novel IGH genes and alleles, and SVs collectively spanning >500 Kb. Using the AIRR-seq data to profile the expressed IgM and IgG repertoire, we directly test for effects of IGH variants on IGHV, IGHD and IGHJ gene usage frequencies. We show that the usage of genes in the IgM and IgG repertoires is associated with IGH germline polymorphism. Strikingly, for a subset of genes, IGH variants alone explain a large fraction of usage variation across individuals and are strongly linked to IGH coding region changes. Finally, we show that IGH gene usage variants are enriched in regulatory elements involved in V(D)J recombination and overlap SNVs previously associated to human phenotypes, offering insight into the underlying mechanisms linking germline variants to gene usage, and highlighting potential pathways from disease risk variant to phenotype. Our results clearly demonstrate that genetics plays a critical role in shaping an individual's Ab repertoire, which will be necessary to understand further in the context of human disease prevention and Ab-mediated immunity.

## Results

### Paired IGH targeted long-read and antibody repertoire sequencing

In this study, we compiled a dataset consisting of newly generated germline IGH locus long-read sequencing data and newly/previously[18] generated AIRR-seq datasets in 154 healthy individuals (Supplementary Data 1). To our knowledge, this dataset represents the most comprehensive collection of matched full-locus IGH germline genotypes and expressed Ab repertoires. Samples in the cohort ranged in age from 17 to 78 years and included individuals who self-reported as White ($n = 81$), South Asian ($n = 20$), Black or African American ($n = 19$), Hispanic or Latino ($n = 19$), East Asian ($n = 11$), Native Hawaiian or Other Pacific Islander ($n = 1$), American Indian or Alaska Native ($n = 1$), or unknown ($n = 2$).

Using our previously published method[28], we performed probe-based targeted capture and long-read single-molecule, real-time (SMRT) sequencing (Supplementary Table 1 and Supplementary Fig. 1a, b) of the IGHV, IGHD, and IGHJ gene regions (collectively referred to as IGH), spanning roughly ~1.1 Mb from *IGHJ6* to the telomeric end of chromosome 14 (excluding the telomere). DNA used for each sample was isolated from either peripheral blood mononuclear cells (PBMCs) or polymorphonuclear leukocytes (PMNs). PBMCs are composed of 70–90% lymphocytes, with B cells making up only 5–10% of the total number of lymphocytes. As a result, we would not expect DNA derived from individual B cell lineages to make significant contributions to the IGH assemblies. The mean coverage across IGH for all individuals ranged from 2× to 331× (mean = 76×) with a mean read length ranging from 3.5 to 8.9 Kbp (mean = 6.4 Kbp; Supplementary Fig. 1c, d). Similar to our previously published work[28], HiFi reads were aligned to a custom linear IGH reference inclusive of previously resolved insertions and used to generate local haplotype resolved assemblies. The mean total number of assembled bases per individual was 2.3 Mb (range = 0.8–3.3 Mb), close to the expected diploid size of IGH (~2.2 Mb); the number and lengths of assembly contigs varied between Pacific Biosciences platforms (Supplementary Fig. 1e–g). These assemblies were then used to curate IGH gene/allele and variant genotype datasets (see below). In contrast to observations made using lymphoblastoid cell lines[28,45], no V(D)J rearrangements were observed in the assemblies, demonstrating that sequencing reads from recombined B cell-derived DNA did not contribute to the assembly process.

AIRR-seq is a powerful technique for analyzing the diversity and composition of expressed adaptive immune receptors. Within a given B cell during development, a single IGHV, IGHD and IGHJ gene are somatically rearranged at the genome level. These recombined IGHV, IGHD, and IGHJ segments are transcribed and spliced together with a constant (IGHC) gene, which determines the receptor isotype (e.g., IgM or IgG). AIRR-seq molecular protocols allow for the selective sequencing of VDJ receptors through the amplification of cDNA (or rearranged genomic DNA) using primers targeting specific IGHC, IGHJ and/or IGHV genes. In the cohort studied here, AIRR-seq data was generated using two different 5′ rapid amplification of complementary DNA ends (5′ RACE) protocols on total RNA isolated from PBMCs collected from 107 individuals. For the remaining 47 individuals, previously generated PBMC derived AIRR-seq data for IgM and IgG was utilized[18]. A standardized workflow was developed to process datasets generated using different protocols and sequencing methods (Methods). Similar sequences with the exact junction length, IGHV and IGHJ allele were grouped into clones ("Methods"). After processing, a mean of 9,038 B cell clones per repertoire was identified (Supplementary Fig. 2a, b). The frequencies of IGHV, IGHD and IGHJ genes among B cell clones were calculated (i.e., gene usage after collapsing sequences by clone) for each individual. Together, these datasets allowed us to resolve large SVs and other genetic variants, and perform genetic association analysis with gene usage variation observed in the expressed Ab repertoire.

## Identification of large breakpoint resolved structural variants

A major goal of this study was to generate a high-confidence set of genetic variants and gene alleles in IGH in order to perform downstream genetic association analysis. Previous reports have demonstrated that SVs are common in IGH, resulting in large insertions, deletions, duplications and complex events[25,27–29,46]. The presence of unresolved SVs can impact the accuracy of variant detection and genotyping. Thus, a key first step in the creation of genotype call sets was to breakpoint resolve and genotype SVs (Fig. 1a–c and Supplementary Fig. 3), which allowed us to account for SVs in determining homozygous, heterozygous, and hemizygous genotypes (Supplementary Fig. 4) across all surveyed variants in the locus.

We utilized our previously published tool, IGenotyper, to generate haplotype-resolved assemblies. We then used these contigs in conjunction with haplotype-specific HiFi reads to create a manually curated genotype call set for large SVs (>9 Kbp; Fig. 1b) within 8

regions of IGH, excluding genotypes in samples that were not supported by haplotype-specific HiFi reads. The eight resolved SV regions (Fig. 1a), 3 of which had overlapping coordinates, were characterized as deletions (*n* = 3), a complex SV (*n* = 1), a duplication (*n* = 1), and multi-allelic SVs (mSV; *n* = 3). Similar to other genetic variant types (e.g., SNVs), an SV allele is defined as an alternative sequence/haplotype relative to the reference. All 8 large SVs altered gene copy number. Four of these regions represented SV hotspots with >2 alleles (Supplementary Data 2), defined by variation in gene copy number. The three mSVs contained 3, 5, and 12 alleles and the duplication contained 3 alleles (Fig. 1a, c). In addition to the SV alleles described in Watson et al.[25],14 new SV alleles were breakpoint resolved, many of which were supported by previous AIRR-seq analysis[26,27,47]. Detailed descriptions of these SVs are provided in the Supplementary Material.

The SV allele frequencies ranged from 0.01 to 0.73 (Fig. 1c). On average across our cohort, relative to the reference assembly used in

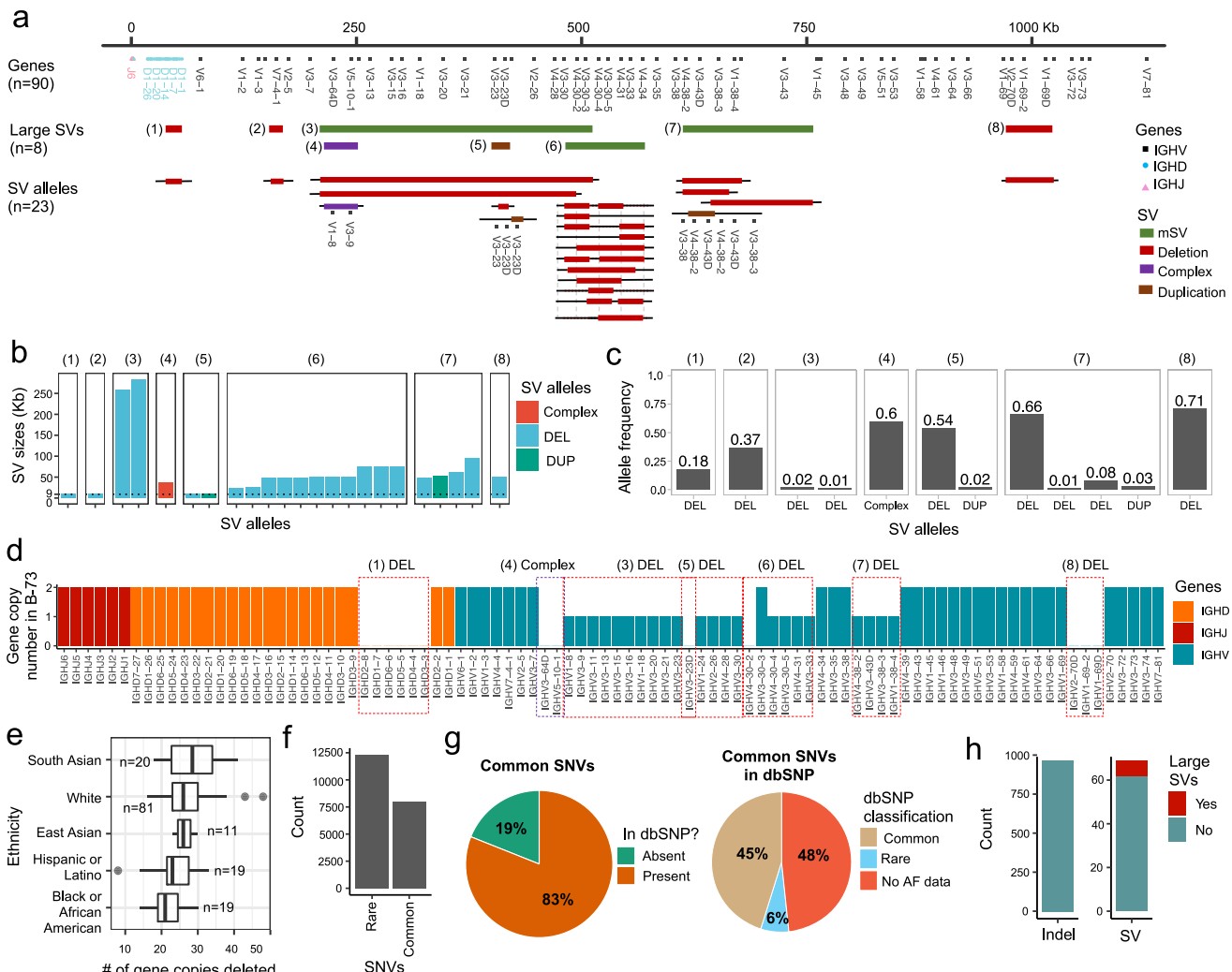

**Fig. 1 | IGH genetic variation identified by long-read sequencing in a cohort of 154 individuals. a** Map of the IGH locus with annotation tracks shown in the following order (top to bottom): joining (IGHJ), diversity (IGHD) and variable (IGHV) genes, structural variants (SV) and SV alleles. The numbers assigned to each SV serve as a unique identifier, distinguishing one SV from another. These numbers are consistently referenced throughout the figures to identify each SV. **b** The sizes for each SV allele. All SVs have at least 1 allele greater than 9 Kbp (black dotted line, y axis). **c** The frequency of alleles for each SV. The allele frequencies for mSV/(6) is not shown. **d** Diploid gene copy number of genes in an individual carrying multiple homozygous and hemizygous deletions; deleted genes are indicated (red boxes;

*n* = 36). **e** Boxplots showing the number of genes deleted for every individual in the cohort grouped by self-reported ethnicity; whiskers and boxes represent the minimum, the maximum, the median, and the first and third quartiles, with outliers plotted as points. **f** Number of characterized SNVs with a minor allele frequency ≥0.05 (common) and <0.05 (rare). **g** Number of common SNVs identified in the study cohort present/absent in dbSNP. A large proportion (54%) of common SNVs identified here using long-read sequencing were missing, defined as rare (6%), or had no allele frequency data in dbSNP (48%). **h** The total count of indels (2–49 bps) and SVs identified (≥50 bps). SV structural variant, IGHV IGH variable, IGHD IGH diversity, IGHJ IGH joining, mSV multi-allelic structural variant, DEL deletion.

our analysis, we found that each individual carried 5.5 large SVs, resulting in homozygous loss of 6.7 genes (range = 0–17), 26.11 gene alleles (range = 14–48; Fig. 1d), and deleted diploid bases summing to 257 Kbp (range = 49–493 Kbp). The observed number of genes and bases deleted within individuals varied by self-reported ethnicity (Fig. 1e). In total, 33 out of 54 IGHV and 6 out of 26 IGHD genes were deleted in 1 or more of the SVs identified in at least one individual (Fig. 1a).

## Long-read sequencing identifies SNVs, indels, and smaller SVs within IGH

SNVs and indels are difficult to characterize within segmental duplications and SVs. Here, we used haplotype-resolved assemblies to more accurately detect and genotype SNVs. In total, we identified 20,510 SNVs in one or more individuals, of which 7980 (39%) were common, defined by a minor allele frequency (MAF) ≥ 0.05 (Fig. 1f). While the majority (97%) of all non-redundant SNVs were in non-coding regions, 472, 103, and 40 SNVs were within exons, introns, and recombination signal sequences (RSS), respectively. Interestingly, SNVs within these genomic features were non-uniformly distributed across IGHV genes (Supplementary Fig. 5). For example, while the mean number of SNVs in IGHV gene RSS was 0.68, several genes, including *IGHV3-21* and *IGHV3-66* had 7 and 5 SNVs in their RSS, respectively. Similarly, the mean number of SNVs across IGHV introns was 1.7, but *IGHV3-23*, *IGHV4-39* and *IGHV7-81* had 9, 8, and 8 intronic SNVs, respectively.

Based on earlier reports of elevated numbers of SNVs in the IGH locus[25], we hypothesized that many of the SNVs identified in this cohort would be novel. Indeed, a total of 4625 (23%) SNVs were not cataloged in dbSNP (release 153), including 1513 (19%) common SNVs (Fig. 1g). Of the total SNVs not in dbSNP, 2393 (59%) were within SVs. Even though a large portion of common SNVs were in dbSNP, we found that 3126 (48%) of the common SNVs had no allele frequency data and 418 (6%) were labeled as rare variants (Fig. 1g). Thus, in total, 63% (5057) of common SNVs identified in our cohort were either missing from dbSNP or are lacking accurate genotype information.

The incomplete and inaccurate genotype frequency information available in dbSNP for IGH is likely in part caused by the prevalence of large SVs in the region, which have hindered the analysis of standard high-throughput genotyping approaches. This is supported directly in our data, as 3406 (43%) of the common SNVs we identified reside within SVs. Here, since SNVs were detected by aligning both haplotype assemblies to the reference, SNVs overlapping heterozygous deletions were simultaneously detected and genotyped as hemizygous (Supplementary Fig. 4). Hemizygous SNVs are often genotyped as homozygous when using short-read and/or microarray data and are excluded from studies due to a departure from Mendelian inheritance and Hardy-Weinberg equilibrium[48]. For 2136 (27%) common SNVs, we observed that the frequency of hemizygous individuals was greater than individuals with both chromosomes present (Supplementary Fig. 4c). Critically, analysis of SNVs within the complex SVs we identified was possible due to long-read assemblies, highlighting the added utility of long-read data in IGH beyond assembly and SV detection.

In addition to SNVs and large SVs, we identified indels (2–49 bp) and small non-coding SVs (50 bp–9 Kbp) using haplotype-resolved assemblies and validated these using mapped HiFi reads (Fig. 1h). In total, 966 indels and 71 small SVs were detected, including expansions and contractions of tandem repeats, mobile element insertions and complex events. We additionally observed highly polymorphic indels and SVs (Supplementary Fig. 6). For example, a tandem repeat with a motif length of 86 bp 5 Kbp upstream of *IGHV3-20* contained 7 tandem repeat alleles ranging in motif copies from 3 to 9 (Supplementary Fig. 6a). Another example includes a complex SV between *IGHV1-2* and *IGHV1-3* with three SV alleles containing multiple copies of a tandem repeat with low sequence matches between motif copies (Supplementary Fig. 6b). An alignment between the 3 SV alleles contains

multiple mismatches including base differences, insertions, and deletions.

## Identification of novel IGH gene alleles using long-read sequencing

Analysis of AIRR-seq data critically relies on the assignment of AIRR-seq reads to specific IGHV, IGHD, and IGHJ gene alleles using existing germline databases. Accurate assignments of reads to gene alleles is used for analyzing a variety of Ab repertoire features, including gene usage and somatic hypermutation. In order to obtain a more complete allele database, we used haplotype-resolved assemblies to annotate additional undocumented novel alleles, defined as alleles absent from the ImMunoGeneTics Information System (IMGT; imgt.org) germline database. In total, we identified 125 IGHV and 5 IGHD high-confidence putative novel alleles (Supplementary Fig. 7), conservatively defined as alleles with exact matches to 10 or more HiFi reads, or identified in two or more individuals (Supplementary Data 3). Of these 125 IGHV alleles, 72 (58%) were found in at least 2 individuals; 23 (18%) and 9 (7%) were found in at least 5 and 10 individuals, respectively; the remaining 53 alleles were found in only one sample, but were supported by ≥10 HiFi reads. Of the 5 novel IGHD alleles, 4 were found in at least 2 individuals and 3 were found in 14 or more individuals. In total, the discovery of 125 and 5 novel IGHV and IGHD alleles represents a 37 and 11% increase in the number of IMGT-documented IGHV and IGHD F/ORF alleles, respectively.

## Gene usage in the expressed antibody repertoire is strongly associated with common IGH variants

Across the genome, genetic variation has consistently been associated with molecular phenotypes such as gene expression and splicing[49]. Performing such analysis on repetitive and SV dense loci such as IGH has been limited by the use of short-read or microarray derived variants. Here, to determine if the long-read sequencing derived genetic variants described above impact the expressed Ab repertoire, we used a quantitative trait locus (QTL) framework (see Materials and Methods) to test if gene usage in the naive (IgM) and antigen-experienced (IgG) repertoire was associated with variant genotypes. The clonal gene usage for 50, 25, and 6 IGHV, IGHD and IGHJ genes, respectively, was tested against all common genetic variants (7042 SNVs, 223 indels, 32 SVs) including SV alleles at 6 of the 8 large (>9 Kbp) SV regions (Fig. 2, Supplementary Fig. 8). In total, across the IgM and IgG repertoires, a collective set of 4380 unique variants (4310 SNVs, 58 indels and 12 SVs) were statistically associated (after Bonferroni multiple-testing correction, $P < 9.2e−6$) with gene usage changes in 40 (80%), 20 (80%), and 4 (66%) unique IGHV, IGHD and IGHJ genes (Table 1), with the majority of associations overlapping between IgM and IgG subsets (Supplementary Fig. 8). Summary data for each gene analyzed in our dataset is provided in Supplementary Data 4 for IgM and IgG. This includes: (1) the number of gene usage QTL (guQTL) variants identified that pass multiple-testing correction; (2) the −log10 P value of the lead guQTL, defined as the variant with the lowest P value; (3) lead guQTL variant type (SNV, indel, SV); (4) the variance explained by the lead guQTL; and (5) the mean fold change in usage between the reference and alternate genotypes. Given the gene usage correlation and high guQTL overlap between IgM and IgG (Supplementary Fig. 9), and the fact that gene usage is a product of V(D)J recombination, we focus on the IgM repertoire in the following results sections.

Given the extent of SVs that alter gene copy number within IGH, we expected to observe effects of large SVs on gene usage. Within the IgM repertoire, there were 5 IGHD genes and 6 IGHV genes that resided within SV regions, and for which the lead guQTL variant was the SV itself or a variant in high LD with the SV ($r > 0.9$; Fig. 2b). These SV associations explained between ~20% and >77% of the variation in IgM usage observed for associated genes (Fig. 2b). As an example, we highlight the association between *IGHV3-64D* usage and a complex SV

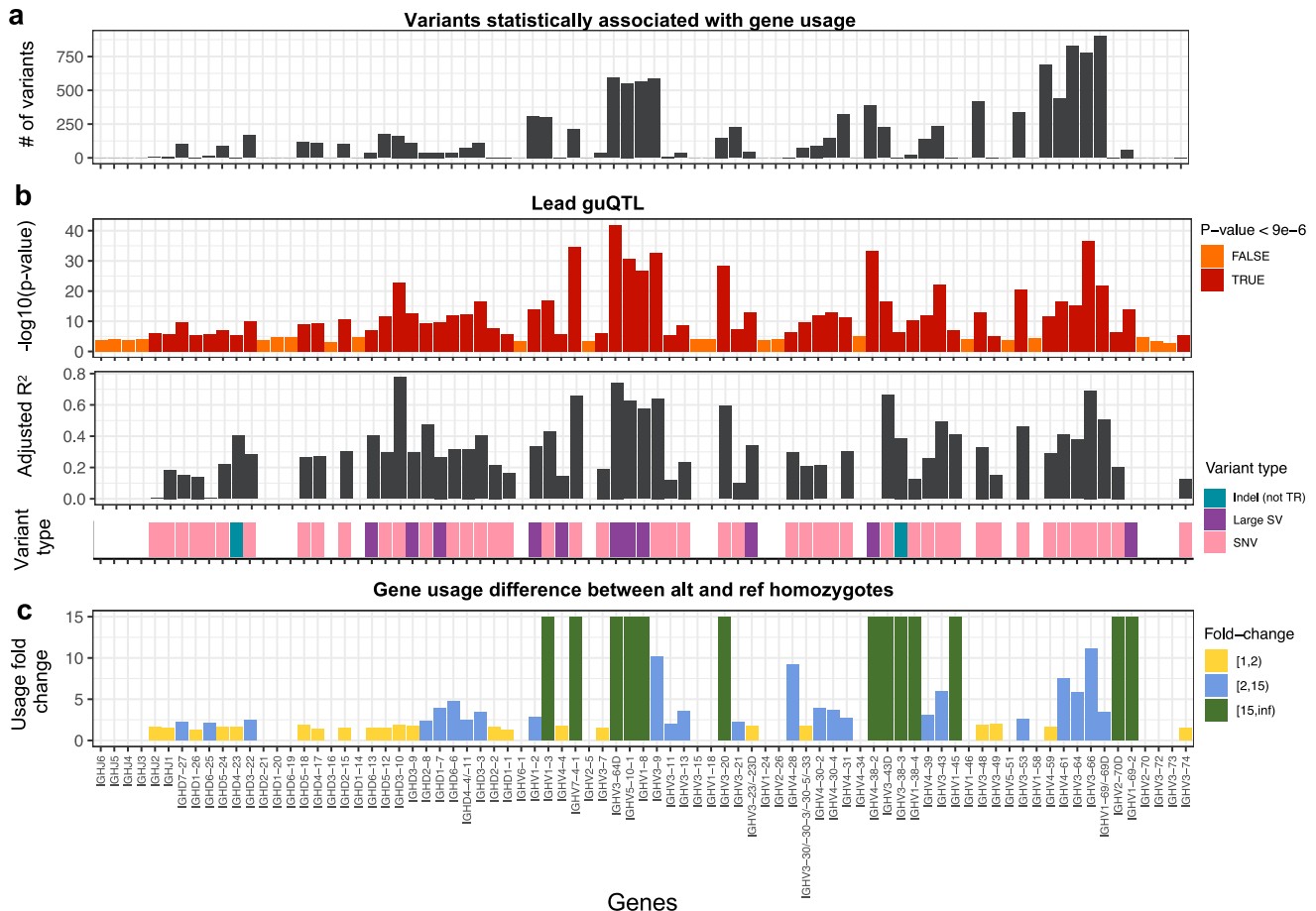

**Fig. 2 | IGH variants impact gene usage in the IgM repertoire.** Per gene (x axis, all panels) statistics from guQTL analysis (ANOVA and linear regression) in the IgM repertoire, including: **a** the number of associated variants (P < 9e−6 threshold after Bonferroni correction); **b** the (i) −log10(P value) of the lead guQTL, (ii) adjusted R² for variance in gene usage explained by the lead guQTL and (iii) the variant type for the lead guQTL; and **c** the fold change in gene usage between genotypes at the lead guQTL. Summary statistics are provided in Supplementary Data 4.

(P = 1.46e−42; Fig. 2b), which alters the genomic copy number of 4 functional IGHV genes (*IGHV3-64D*, *IGHV5-10-1*, *IGHV1-8*, and *IGHV3-9*) from 0 to 2 diploid copies (Fig. 1a). The impact on gene usage of this SV was as expected, following an additive model in which individuals with zero copies of a given gene had the lowest mean usage (in this case 0%), whereas individuals with 2 diploid copies of a given gene had the highest mean usage, and heterozygotes showed intermediate usage. Other large deletions followed a similar pattern. The deletion spanning the genes *IGHD2-8* to *IGHD3-3* was associated with the usage of six IGHD genes (Fig. 3a), five of which reside within the deletion (*IGHD2-8*, *IGHD1-7*, *IGHD6-6*, *IGHD4-11/4-4*, and *IGHD3-3*; Fig. 3a); these results were consistent with those noted previously[42]. Due to low frequency, the largest mSV alleles (Supplementary Fig. 3a and Fig. 1a), which resulted in deletion of 16 IGHV genes were not tested; however, we observed empirically that the 7 individuals carrying either one of these large deletions had decreased usage across 15 out of the 16 genes (Supplementary Fig. 10). In addition to SVs that resulted in gene

deletions, we also noted an association with the duplication characterized for the *IGHV3-23/D* genes, at which we tested for effects of copy number genotypes between 2 to 4 diploid copies. Again, this effect was consistent with an additive contribution of gene copy number, with mean usage increasing incrementally from 7.4% in individuals with 2 copies, to ~13% in individuals with 4 copies (Fig. 3b); individuals carrying the rare 3-copy haplotype (Supplementary Figs. 3d and 11) were excluded from this analysis.

We additionally identified 3 IGHD genes (*IGHD6-13*, *IGHD3-9* and *IGHD3-10*) and 2 IGHV genes (*IGHV1-2 and IGHV4-4*) that were associated with SVs or a variant in high linkage disequilibrium (LD, r² > 0.9) with a SV, although the copy number of these genes was not directly altered (Supplementary Data 4). The deletion spanning the IGHD genes mentioned above was the lead variant associated with *IGHD3-10* usage, even though the gene is ~3 Kbp away from the deletion. Contrary to genes residing within the deletion, the mean usage of *IGHD3-10* increased from 10 to 19% in individuals with the deletion on both haplotypes (Supplementary Fig. 12), suggesting that the deletion modulated the usage of these genes through *cis*-regulatory mechanisms[50,51]. Interestingly, usage of the gene *IGHV1-69-2*, which resides within a deletion SV, was associated with a secondary SV, located ~322 Kb away. However, given the low usage of *IGHV1-69-2*, deeper repertoire sequencing will likely be needed to tease out the effect of both SVs.

We next focused on the 42 genes (IGHJ, n = 2; IGHD, n = 12; IGHV, n = 28) for which the lead guQTL was not an SV. The lead guQTLs associated with 40 of these genes were SNVs, and the remaining 2 were

**Table 1 | Number of variants and genes identified by guQTL analysis (ANOVA and linear regression; P < 9.2e−6)**

| Repertoire | # of variants | | | # of genes | | |
|---|---|---|---|---|---|---|
| | SNVs | Indels | SVs | IGHJ | IGHD | IGHV |
| IgM | 3967 | 50 | 8 | 2 (33%) | 20 (80%) | 37 (74%) |
| IgG | 3675 | 36 | 11 | 3 (50%) | 14 (56%) | 33 (66%) |
| IgM + IgG | 4310 | 58 | 12 | 4 (66%) | 20 (80%) | 40 (80%) |

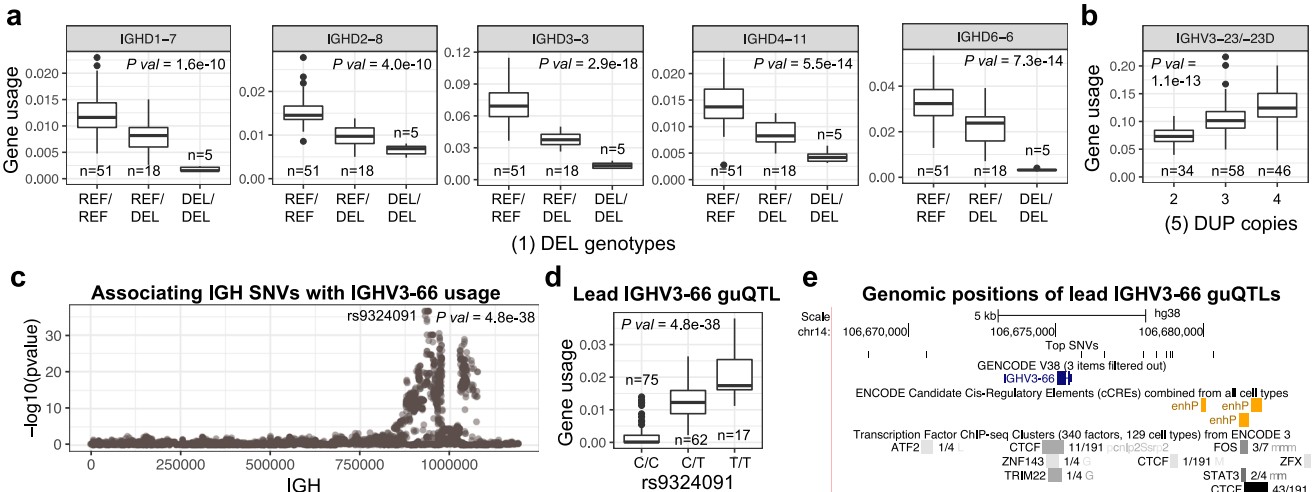

**Fig. 3 | Associations of IGH SVs and SNVs with gene usage in the IgM repertoire. a** Gene usage for genes within the IGHD gene region deletion (see Fig. 1a, b). Individuals homozygous for the deletion ("DEL/DEL") use those genes at lower frequencies than the rest of the cohort. **b** Gene usage for *IGHV3-23/-23D* in individuals partitioned by gene copy number (see Fig. 1a, b). Individuals carrying more gene copies use these genes at higher frequencies. **c** SNVs associated with the usage of *IGHV3-66* using linear regression. The Manhattan plot shows the −log10(*P* value) for all SNVs in the IGH locus tested for *IGHV3-66*; there are 10 lead SNVs/guQTLs with the same *P* value (*P* value = 4.8e−38). Dark red SNVs are those SNVs that passed Bonferroni correction (*P* value < 9e−6). **d** *IGHV3-66* usage in individuals partitioned by genotypes at 1 of the 10 lead guQTLs. **e** Genomic localization (hg38; GRCh38) of lead guQTLs (top track) relative to *IGHV3-66*, as well as cCRE and TF locations (middle and bottom tracks). Genomic map was made using the UCSC Genome Browser (https://genome.ucsc.edu/). Boxplots display the median, 25th percentile, 75th percentile, and whiskers that extend up to 1.5 times the inter-quartile range (IQR) from the respective percentiles. Data points outside the whiskers are also plotted. DUP duplication.

indels; although we identified the presence of smaller SVs and tandem repeats in our dataset, none of these were found to be lead variants in our analysis. For 38 of the genes, we identified between 2 and 900 guQTLs (Fig. 2a), reflecting local haplotype structure. In some cases, an SNV or indel was the lead guQTL for genes residing within SVs indicating that multiple variant types need to be taken into account to fully model the genetic effects on usage (see below). Similar to SVs, lead guQTLs that were SNVs or indels explained a significant fraction of usage variation, in some cases up to 69% (range, $R^2$ = 0.003−0.69; mean = 0.29), exhibiting large usage differences between genotype groups (Fig. 2b). The lead guQTLs for all 42 genes resided within non-coding regions. The median genomic distance between inter-genic guQTLs and their associated genes was 5.1 Kbp (min = 13 bp, max = 1.1 Mbp).

The SNV-driven guQTL in this dataset with the lowest *P* value was for *IGHV3-66* (*P* value = 2.86e−37; Fig. 3c−e). In total, there were 776 SNVs associated with the usage of *IGHV3-66* (Fig. 3c). These included 10 lead SNVs in perfect LD ($r^2$ = 1), spanning a region of 11.6 Kbp surrounding the gene, which explained ~69% of variation in usage, representing a mean fold-change in usage of 11.2-fold between the two homozygous genotypes (Fig. 3d, e).

**Conditional analysis identifies multiple variants associated with the usage of single genes**
Previous eQTL studies have demonstrated that multiple independent variants can influence gene expression[49]. Here, we hypothesized that the usage of individual genes could be affected by multiple variants, such as multiple SNVs, or a combination of variant types. To test this, we performed a conditional analysis by running an additional guQTL test in individuals homozygous for either the reference or alternate allele for the lead guQTL variant of all genes. Out of the 59 genes statistically associated (*P* < 9.2e−6; Table 1) with gene usage in the IgM repertoire, 55 genes were tested for additional associations. The 4 genes not tested had fewer than 50 individuals with homozygous reference or alternate allele genotypes. From this analysis, we identified 14 genes with significant secondary/conditional guQTLs (Supplementary Data 5). For 12 of these 14 genes, the lead guQTL and

secondary guQTL were 2 SNVs, and for the remaining 2 genes, this analysis revealed combined effects of an SV (lead guQTL) and SNV (secondary guQTL). The mean genomic distance between the lead and secondary guQTL variants was 36.2 Kbp (range = 1.7−161.4 Kbp). Here, we present *IGHV1-2* (Fig. 4) and *IGHV3-66* (Supplementary Fig. 13) as examples of genes associated with 2 independent variants. Data for all genes is provided in Supplementary Data 5.

For *IGHV1-2*, the lead guQTL was an SV ~31 Kb away from *IGHV1-2* (Fig. 4a), which involved the deletion of *IGHV7-4-1*. Individuals homozygous for the deletion used *IGHV1-2* at a 2.8-fold higher rate than individuals homozygous for the reference allele (Fig. 4b). Conditioning on individuals without the deletion, identified 35 SNVs additionally associated with the usage of *IGHV1-2* (Fig. 4c). Of these individuals, heterozygotes for the secondary lead conditional guQTL used *IGHV1-2* (Fig. 4d) at a level (mean usage = 3.8%) similar to those with a deletion in both haplotypes (mean usage = 4.2%). Sequencing data from het-erozygotes at the lead conditional guQTL were inspected manually to confirm that *IGHV7-4-1* deletions were not present in these individuals.

For *IGHV3-66*, the lead guQTL was an SNV. Individuals homo-zygous for the reference and alternate allele had a mean usage of 0.19 and 2.14%, respectively (Supplementary Fig. 13a). By conditioning on this variant, considering only individuals homozygous for the refer-ence allele, a total of 438 additional SNVs were significantly associated with *IGHV3-66* usage (Supplementary Fig. 13b). At the SNV with the lowest P value from this analysis, only reference allele homozygotes and heterozygotes were observed. In heterozygotes, the mean usage was 0.006% compared to 0.0003% in homozygotes, with many indi-viduals in the homozygote group exhibiting 0% usage (Supplementary Fig. 13c). Thus, based on this conditional guQTL analysis, variation in *IGHV3-66* usage can be further explained even in individuals with relatively low usage.

**Gene by guQTL network analysis reveals that the usage of multiple genes is associated with overlapping sets of variants**
In addition to discovering multiple variants associated with the usage of a single gene, our guQTL association analyses also identified single variants associated with the usage of multiple genes. This was

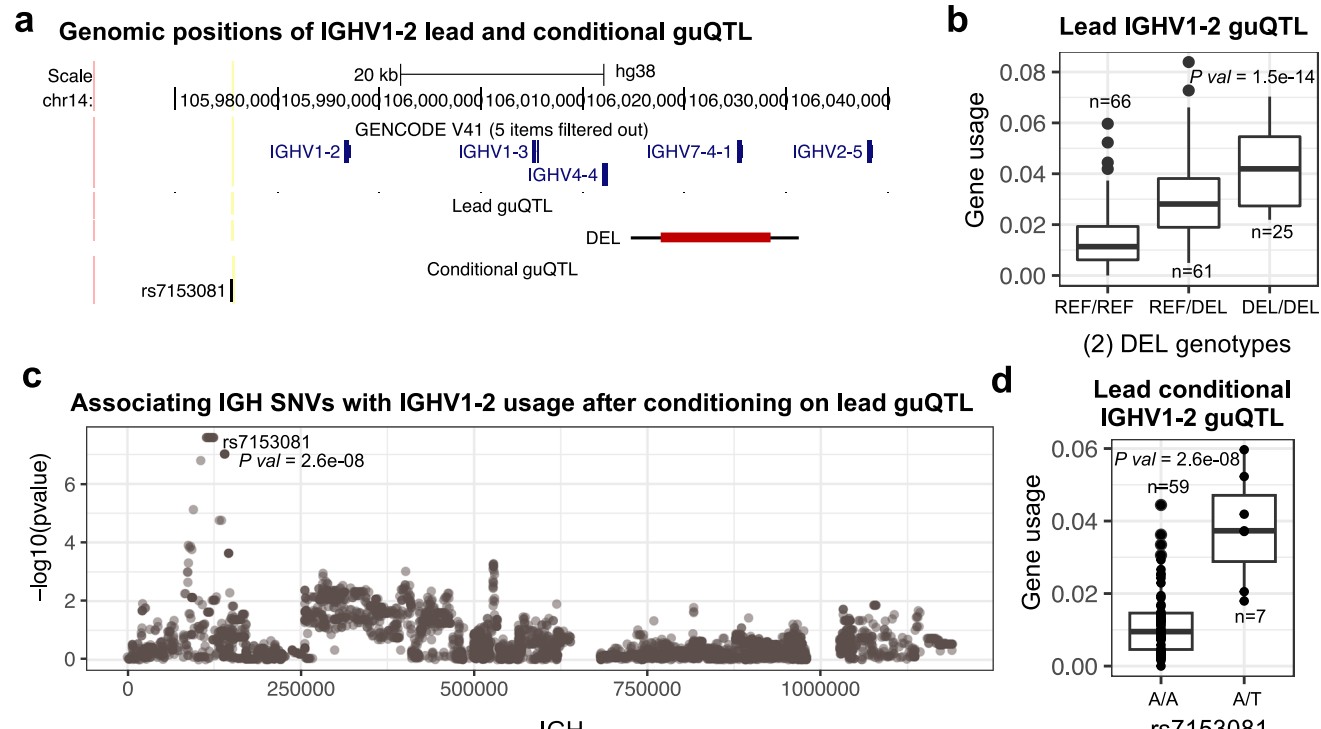

**Fig. 4 | Example of additional variants associated with gene usage after conditioning on lead guQTL. a** Map showing positions of the lead and conditional guQTLs for *IGHV1-2* (bottom tracks). **b** *IGHV1-2 usage* in individuals partitioned by SV genotype; individuals homozygous for the *IGHV7-4-1* deletion have greater *IGHV1-2* usage on average. **c** Manhattan plot showing the statistical significance of all SNVs tested for secondary effects on *IGHV1-2* gene usage using linear regression (red indicates Bonferroni corrected significant SNVs), after conditioning on genotype at the *IGHV7-4-1* SV. **d** *IGHV1-2* usage among individuals of the "REF/REF" *IGHV7-4-1* SV genotype (**b**), partitioned by genotype at the secondary guQTL (**c**). Boxplots display the median, 25th percentile, 75th percentile, and whiskers that extend up to 1.5 times the inter-quartile range (IQR) from the respective percentiles. Data points outside the whiskers are also plotted.

intriguing as V(D)J recombination studies in animal models have demonstrated the coordinated selection of genes through the same regulatory elements[32,52]. In mice, IG V genes reside in topologically associating domains (TADs) and disruption of regulatory elements within the IG loci has been shown to cause altered gene usage within these domains[53-55]. Given this, we further assessed coordinated genetic signals involving sets of multiple variants and genes. We found that 2,607 (66%) guQTL variants were associated ($P < 9.2e−6$) with >1 gene (Fig. 5a). We reasoned that this could have multiple underlying causes: (1) the SNV is tagging an SV overlapping multiple genes; (2) the SNV is tagging multiple causative regulatory SNVs; (3) the SNV is overlapping a regulatory element controlling multiple genes; or (4) a combination of any of the prior explanations.

To determine the set of guQTL genes with the same set of guQTL variants, we created a network with genes as nodes and edges connecting genes associated with the same guQTL SNVs (Supplementary Fig. 14). The weight of the edges corresponded to the number of guQTL SNVs connecting two genes. A total of 23 cliques (subgraphs in which all genes are connected) were identified with edge weights >2 (i.e., more than 2 SNVs connecting 2 genes; Supplementary Fig. 15). These 23 cliques included a total of 16 IGHD and 29 IGHV genes, with the number of genes per clique ranging from 2 to 9. Out of the 23 cliques, 10 were primarily composed of genes within SVs.

We also identified cliques made up primarily of genes outside of SVs (Fig. 5b). For example, the SNV shown in Fig. 5c was associated with the usage of 7 genes, *IGHV4-31, IGHV3-S3, IGHV4-59, IGHV4-61, IGHV3-64, IGHV3-66* and *IGHV1-69/-69D*; this variant was located -120 Kbp away from the nearest SV, and exhibited low LD with the SV ($r^2 = 0.09$). Interestingly, gene usage patterns associated with this SNV were either negatively or positively correlated depending on the gene (Fig. 5d). Individuals homozygous for the reference allele had higher usage of

*IGHV4-31, IGHV3-S3, IGHV4-61* and *IGHV1-69/-69D* and lower usage for the remaining genes. In summary, we show that the usage of specific sets of genes in the repertoire are associated with the same sets of variants, indicating the potential for complex and coordinated regulatory mechanisms.

### Variants associated with gene usage variation are enriched in regulatory regions involved in V(D)J recombination

Large-scale studies using expression, epigenomic and disease or trait-associated variant datasets have identified non-coding variants in regulatory elements linked to their phenotypes of interest[49,56-58]. Specific to V(D)J recombination, recombination signal sequences (RSS) are sequence motifs in IG and T cell receptor non-coding regions used by RAG1/RAG2 proteins to direct double-strand DNA breaks and initiate somatic recombination[59]. Additionally, CCCTF-binding factor (CTCF) and cohesin binding has been shown to regulate locus contraction and recombination in IGH[60-62]. We therefore hypothesized that variants might modulate gene usage through regulatory elements such as CTCF-binding sites. To test this, we tested for the enrichment of guQTL SNVs within ENCODE Registry candidate *cis*-Regulatory Elements (cCREs) (Fig. 6a). The cCRES were split into 9 classifications: (1) CTCF-only and CTCF-bound, (2) proximal enhancer-like and CTCF-bound, (3) proximal enhancer-like, (4) DNase and H3K4me3, (5) promoter-like, (6) distal enhancer-like, (7) distal enhancer-like and CTCF-bound, (8) DNase, H3K4me3, and CTCF-bound, and (9) promoter-like and CTCF-bound. Using a one-sided Fisher exact test, we determined that guQTL SNVs were significantly enriched within CTCF-only and CTCF-bound (Fishers exact, $P = 3.8e−04$) and distal enhancer-like and CTCF-bound ($P = 0.014$). An enrichment in cCREs marked by DNase and H3K4me3 was also observed, but was not statistically significant (Fishers exact, $P = 0.08$). A total of 23 out of 3573 guQTL SNVs tested were within

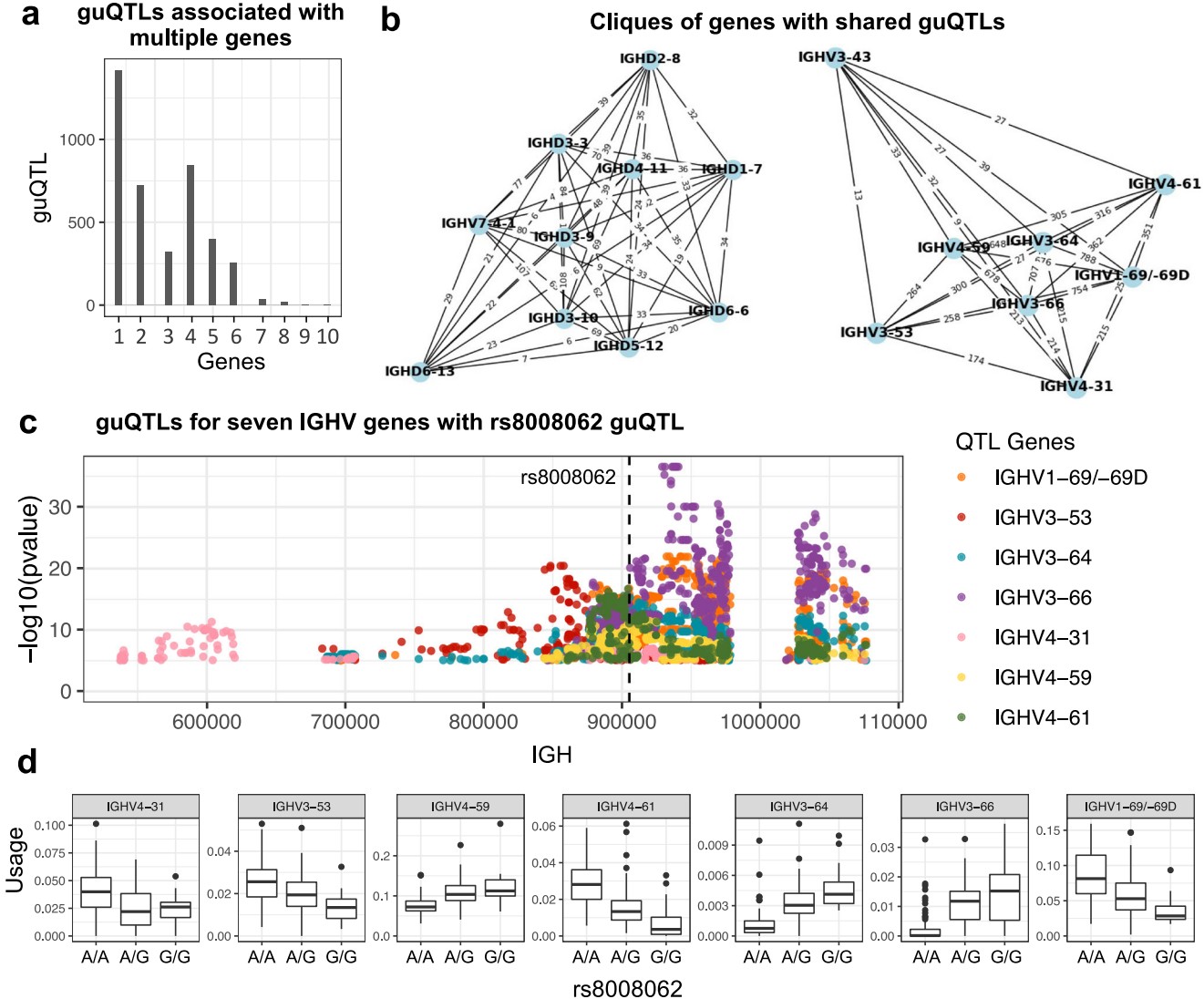

**Fig. 5 | guQTL network analysis reveals coordinated genetic effects on gene usage patterns. a** Bar plot showing the number of SNVs (guQTLs) significantly associated (linear regression; *P* value < 9e−6) with varying numbers of genes (*n* = 1–10); this includes a large number of SNVs that were associated with >1 gene (see Fig. 2). **b** Examples of cliques identified from a comprehensive network of genes and guQTLs (see also Supplementary Figs. 14 and 15), demarcating groups of genes associated with overlapping sets of guQTLs. For each clique, genes are shown as nodes, connected by edges displaying the number of shared guQTLs. **c** Manhattan plot showing statistically significant SNVs (linear regression;

*P* value < 9e−6) associated with the usage of 7 genes; each point is colored by the gene it is associated with. The position of an SNV (rs8008062) associated with all 7 genes is indicated by the dashed line. **d** Boxplots show usage variation for each gene partitioned by genotypes at this SNV. The number of individuals with A/A, A/G and G/G genotypes is 61, 69, and 24, respectively. Boxplots display the median, 25th percentile, 75th percentile, and whiskers that extend up to 1.5 times the inter-quartile range (IQR) from the respective percentiles. Data points outside the whiskers are also plotted.

CTCF-only and CTCF-bound cCRE compared to 2 out of 2419 common non-guQTL SNVs. These 23 SNVs were significantly associated with 3 IGHD genes and 19 IGHV genes and resided within 12 distinct cCREs. Interestingly, 4 SNVs within a CTCF-only and CTCF-bound cCRE (ENCODEAccession: EH38E1747546; chr14:106695880–106696139 (hg38)) were found between *IGHV3-66* and *IGHV1-69* and associated with usage of *IGHV3-53*, *IGHV4-59*, *IGHV3-66*, *IGHV3-64* and *IGHV1-69/-69D*, included in the clique noted above (Fig. 5c, d). Within the DNAse and H3K4me3 cCREs, there were 10 SNVs associated with gene usage for eight and two IGHD and IGHV genes, respectively. H3K4me3 is critical for V(D)J recombination via interaction with RAG2; disruption of the binding between RAG2 and H3K4me3 has been shown in vivo to reduce V(D)J recombination[63].

We additionally compared the enrichment of guQTLs in specific transcription factor binding sites (TFBS) using the ENCODE3 Transcription Factor ChIP-seq binding site dataset (Fig. 6b). A total of 365

TFBS with high normalized ChIP-seq signals were tested. Again, an enrichment of guQTLs in the CTCF binding sites was observed (Fishers exact, *P* = 0.004). Significant enrichments were observed for eight additional TFBSs (*P* < 0.05), including *EED*; the disruption of *Eed* in mice has been shown to affect IGHV gene usage[54]. The fact that SNVs are enriched in sites associated with V(D)J recombination rather than transcription (e.g. promoters and enhancers) provides strong initial support that the guQTLs identified here impact gene usage via effects on V(D)J recombination.

## IGH gene alleles are linked to guQTLs

IGH germline coding variants can directly alter Ab function by modifying antigen binding[23,64,65], and previous studies have demonstrated that specific coding alleles are utilized at different frequencies within the repertoire[23,41]. To assess this more comprehensively in our dataset, we tested for associations between IGH gene alleles and all lead

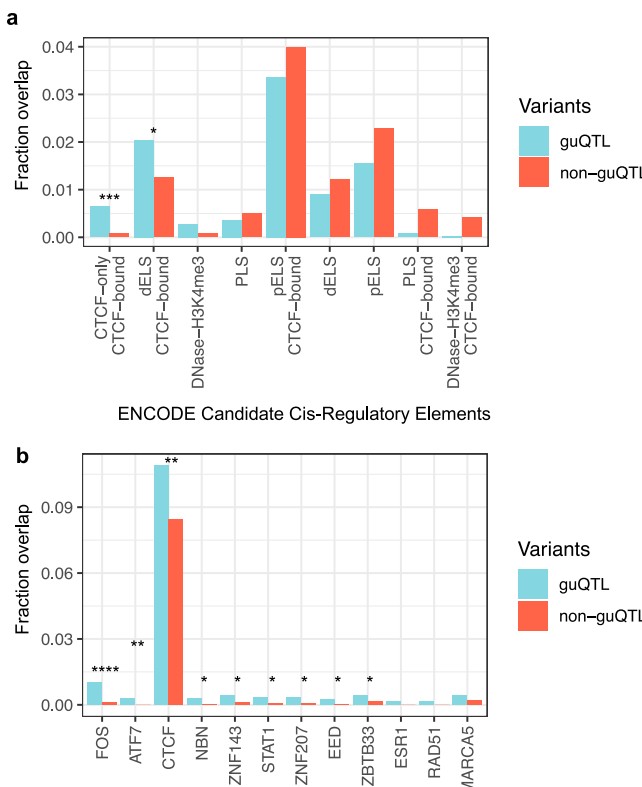

**Fig. 6 | Enrichment of guQTL variants in regulatory elements and transcription factor binding sites involved in V(D)J recombination. a**, **b** Bar plots showing the fraction of guQTL SNVs (*P* value < 9e−6) that overlapped (**a**) ENCODE candidate *cis*-regulatory elements, and (**b**) ENCODE3 TFBS, compared to the overlap observed for the non-guQTL set of variants used in the guQTL analysis. Regulatory elements and TFBS for which statistically significant enrichments were observed are indicated by asterisks: One-side Fisher's Exact Test; *P* value < 0.05; **P* value < 0.005; ***P* value < 0.0005; ****P* value < 0.00005.

guQTLs (Fig. 7). We found that allele frequency distributions at 21 IGHV genes were different based on lead guQTL genotype (Fisher exact test, *P* < 0.05; Supplementary Data 6). The top three genes that exhibited coding allele genotype biases (based on *P* value) between guQTL variant genotype groups were *IGHV3-64* (*P* = 6.9e−57; Fig. 7a), *IGHV3-53* (*P* = 4.4e−54; Fig. 7c), and *IGHV3-66* (*P* = 5.0e−49; Fig. 7c). In the case of *IGHV3-66*, out of the 62 individuals who were homozygous for the reference allele at the lead *IGHV3-66* guQTL, 35 (52%) and 15 (23%) were homozygous and heterozygous, respectively, for the *IGHV3-66*03* allele. In contrast, *IGHV3-66*03* was not observed in any of the individuals homozygous for the alternate allele at this guQTL, which were all homozygous for *IGHV3-66*01*. These results show a direct genetic link between gene usage and coding variation, indicating that both should be considered in future studies investigating germline effects on Ab function.

**Variants linked to disease and other traits overlap guQTLs**
Biased gene usage has consistently been observed in autoimmune and infectious diseases[37,66]. We have argued that one possible explanation for these biases is that they are mediated through genetic variants that influence Ab antigen specificity and/or gene usage[22]. Integrating genome-wide association studies (GWAS) and eQTL datasets has been an effective method for assessing the potential links between genetic variation, function and disease pathology[48,67,68]. Here, we assessed whether IgM and IgG guQTL SNVs were also identified by GWAS (Fig. 8a). In total, across IGH (chr14:105,860,000-107,043,718,

GRCh38) there were 41 SNVs associated with 17 traits/diseases reported in the NHGRI GWAS catalog (*P* < 4e−6). In total, 22 SNVs from 10 independent GWAS performed on 8 diseases/traits overlapped guQTL SNVs[64,69–77]. These included SNVs associated with rheumatic heart disease (RHD) and Kawasaki disease (KD). In both diseases, SNVs were significantly associated with the usage of genes previously implicated by GWAS (*IGHV4-61* for RHD and *IGHV3-66* for KD)[64,69]. In the case of RHD, the risk variant identified in IGH is the strongest genetic association identified to date for this disease[64], and has implicated *IGHV4-61*02* in increased risk. Interestingly, only individuals with the GWAS-guQTL SNV reference allele carried *IGHV4-61*02*, and these individuals had significantly lower *IGHV4-61* usage in IgM and IgG. In both RHD and KD, the usage of additional genes were also associated with the same guQTL SNV. For KD, the SNVs detected in the GWAS were also associated with *IGHV1-69/-69D*, *IGHV3-64* and *IGHV4-61* usage (Fig. 8b). Similar to using expression data to prioritize genes affected by SNVs identified from GWAS, here we show that guQTL-GWAS SNVs are associated with the usage of multiple genes in the Ab repertoire. Additional diseases/traits associated with SNVs identified by both GWAS and our guQTL analysis included the proportion of morphologically activated microglia in the midfrontal cortex, and estradiol levels, which were associated with the usage of *IGHV1-69/-69D* and *IGHV2-70D*, and *IGHV1-8*, *IGHV3-64D*, *IGHV3-9* and *IGHV5-10-1* usage, respectively (Fig. 8c). In both examples, the GWAS SNVs and guQTLs were in strong LD with SVs spanning these respective sets of candidate genes (*r* = 0.51 and *r* = 0.98) suggesting that the observed effects could at least in part be SV mediated.

**Repertoire-wide gene usage profiles are more highly correlated in individuals carrying shared IGH genotypes**
Previous studies in monozygotic twins have shown that gene usage frequencies in genetically identical individuals are more highly correlated than in unrelated individuals[20,21]. We reasoned that such effects could also be observed at the population level by assessing correlations in individuals sharing greater versus fewer IGH guQTL SNV alleles. To assess this, we used allele sharing distance[78,79] (ASD) to group individuals with similar genotypes across IGH and compare the IgM gene usage correlation between groups. Two ASD-based groupings were performed using either (1) the lead guQTL per gene (Fig. 9a), or (2) all guQTLs (Fig. 9b). We tested the latter case as we noted above that multiple variants could influence a single gene, and it has been shown that accounting for a greater number of common variants associated with a given phenotype can explain more variation in that phenotype[80]. Repertoire-wide gene usage correlations between samples were calculated using the Pearson's Correlation coefficient. Using only the lead guQTL variants for each gene, individuals with the most overlapping guQTL genotypes (low ASD) had a higher mean IgM gene usage correlation than those in the group with the highest ASD scores (0.958 vs. 0.943; KS test *P* value < 3.8e−15). The same pattern was observed when using all statistically significant (*P* < 9.2e−6; Table 1) IgM guQTL variants (0.956 vs. 0.943; KS test *P* = 0.008). These results indicated that genetic background makes a contribution to the overall gene usage composition of the repertoire, and expand on previous observations made in twin studies[20,21], by demonstrating that heritable components of the heavy chain repertoire can be directly linked to germline variants in the IGH locus.

**Discussion**
In this study, we show conclusively that IGH genetic polymorphisms influence the composition of the Ab repertoire through impacts on gene usage frequencies. Resolution of complex IGH genetic variants using long-read sequencing identified associations between these variants and gene usage within the IgM and antigen-stimulated (IgG) repertoire. Variants were found to affect the Ab repertoire via (1) SVs that alter IGH gene copy number, including deletions that completely

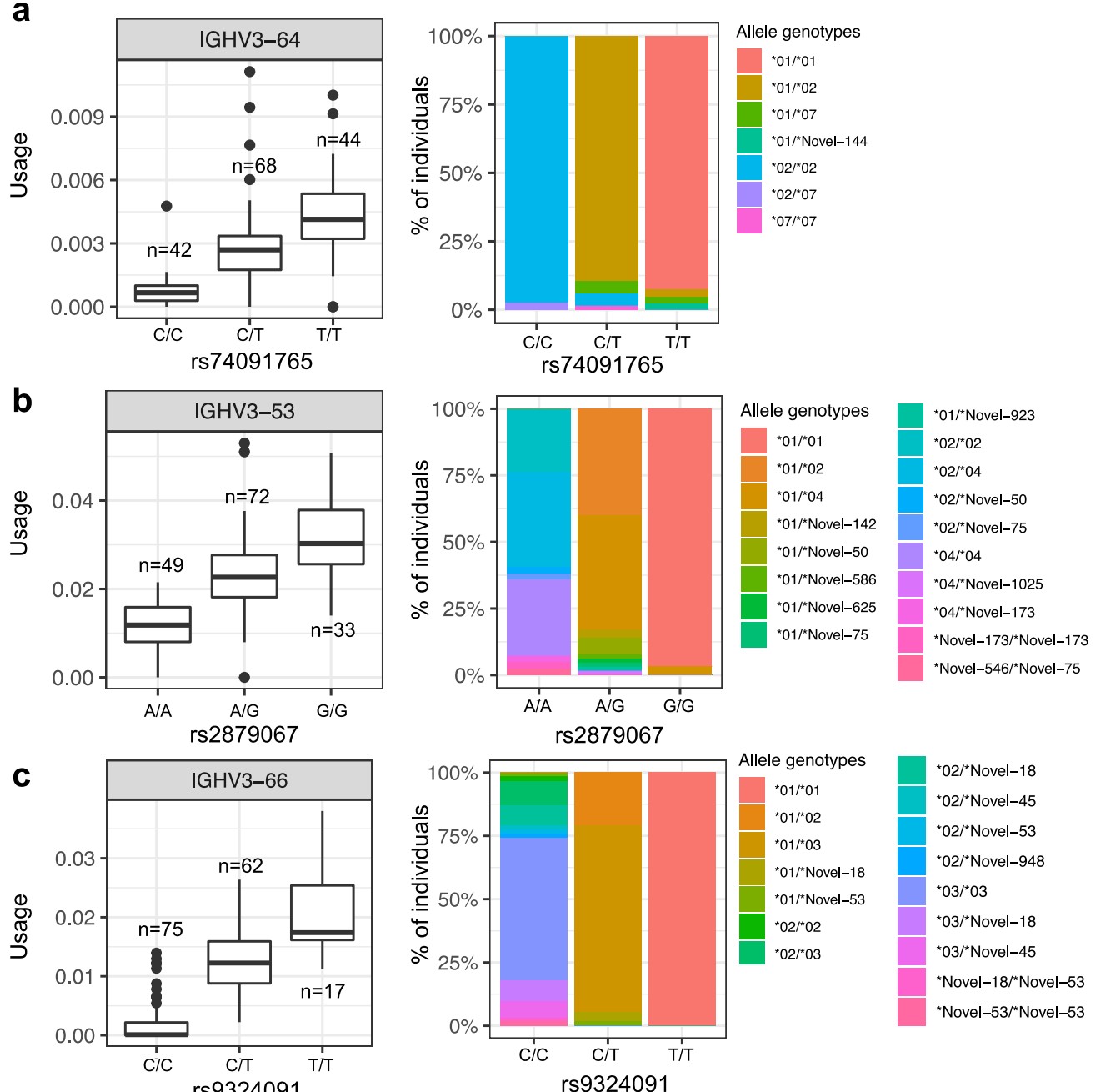

**Fig. 7 | Association between IGHV coding region alleles and lead guQTL genotypes. a–c** For each IGHV gene, the distribution of coding region allele-level genotypes among individuals partitioned by genotype at the lead guQTL for that gene was assessed (Fisher's exact test). For the three genes with the lowest *P* values from this analysis (**a**; *IGHV3-64*, **b**; *IGHV3-53* and **c**; *IGHV3-66*), IgM gene usage (boxplots) and the distributions (stacked bar plots) of the respective coding allele genotypes across individuals partitioned by guQTL genotype are provided. Box-plots display the median, 25th percentile, 75th percentile, and whiskers that extend up to 1.5 times the inter-quartile range (IQR) from the respective percentiles. Data points outside the whiskers are also plotted.

remove genes from the repertoire, as well as through (2) SNVs and indels, including those overlapping regulatory elements and transcription factor binding sites linked to V(D)J recombination. The strength of these associations was substantial, in some cases explaining >70% of variance in usage of particular genes. Building on past observations from twin studies[20,21], we found that repertoire-wide gene usage patterns were more similar in individuals sharing a greater number of genotypes across IGH. Together, these findings (1) advance our basic understanding of repertoire development, illuminating regions of IGH involved in gene regulation, and (2) more broadly represent a paradigm shift towards a model in which the Ab repertoire is formed by both deterministic and stochastic processes. This shift

has critical implications for delineating the function of Abs in disease, with great potential to inform the design and administration of therapeutics and vaccines.

SVs are a hallmark of the IGH locus[25–27,47,81], which was clearly supported by our analysis. We breakpoint resolved 23 SV haplotypes/alleles within 8 different SV loci spanning 542 Kbp of IGH; this included 14 novel SV alleles, and collectively resulted in copy number changes in 6 IGHD genes and 33 IGHV genes, representing 22 and 61% of all IGHD and IGHV genes in IGH, respectively. Critically, our ability to resolve SVs allowed us to more comprehensively detect and genotype SNVs and indels. In total, we identified 20,510 unique SNVs and 966 indels, 7980 and 223 of which were common. A significant fraction of these

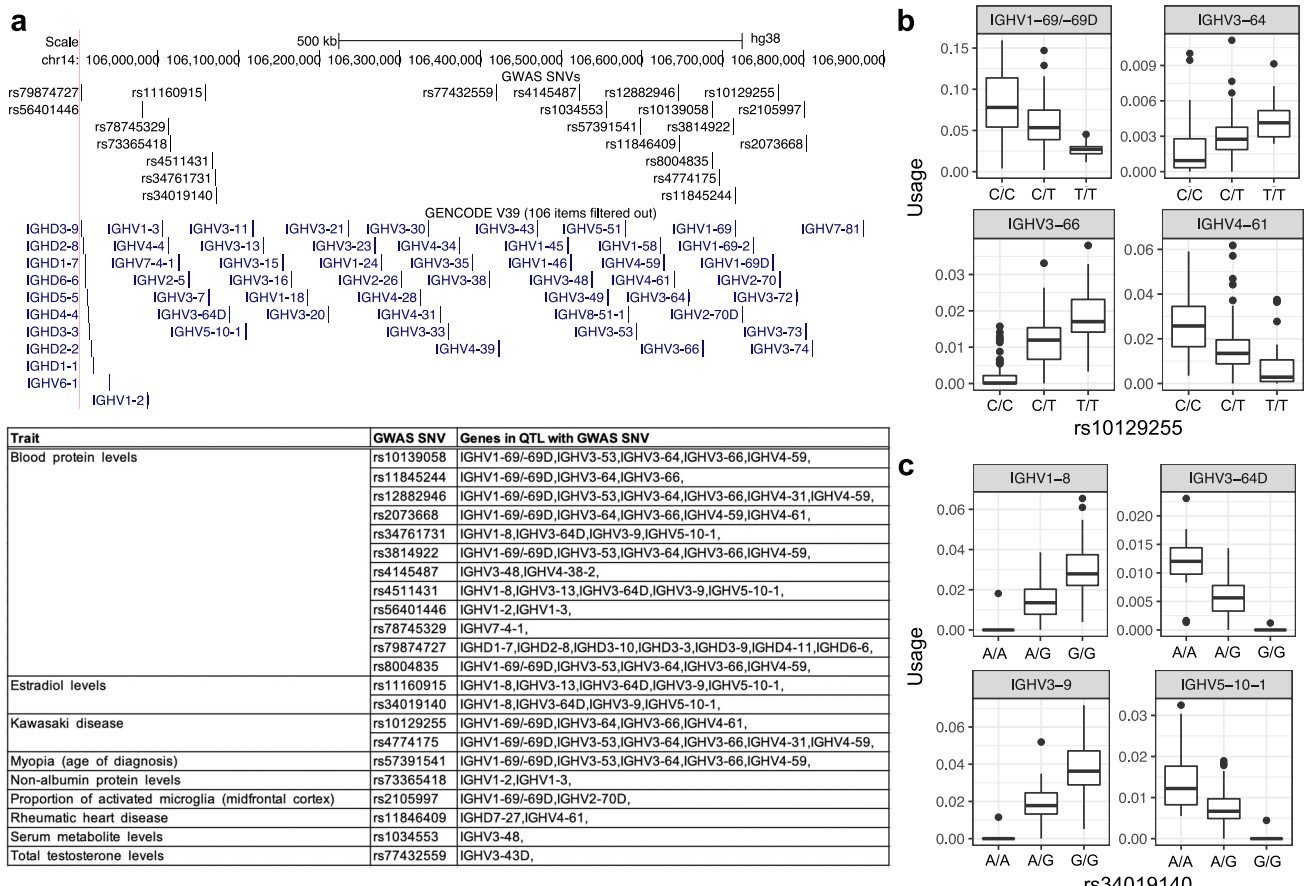

**Fig. 8 | SNVs associated with diseases and other clinical traits are also associated with gene usage variation. a** Map of IGH (GRCh38) showing the positions of SNVs identified by genome-wide association studies (GWAS); positions of F/ORF genes are also provided. For each GWAS SNV found to overlap a guQTL (IgM and IgG) from our dataset, the table provides information on the trait, SNV identifier, and genes for which usage was associated with the GWAS/guQTL SNV. **b**, **c** Boxplots showing gene usage variation for all genes associated with two GWAS SNVs for (**b**) Kawasaki disease and (**c**) estradiol levels. The number of individuals with C/C, C/T, and T/T genotypes for rs10129255 is 67, 67, and 20, respectively. For rs34019140, the numbers are 23, 79, and 52 for A/A, A/G, and G/G genotypes, respectively. Boxplots display the median, 25th percentile, 75th percentile, and whiskers that extend up to 1.5 times the inter-quartile range (IQR) from the respective percentiles. Data points outside the whiskers are also plotted.

overlapped SVs ($n$ = 3406), which we accurately genotyped as hemizygous. Additional novelty was discovered through the annotation of IGH genes, revealing 130 undocumented alleles not currently curated in the germline gene database IMGT[82]. Together, these data hint at the extent of variation that we have yet to describe in this complex locus, and bolster previous concerns that past genetic studies have overlooked IGH variants[28,31,45]. A major outcome of this study is that these data can start to be used to augment existing resources and databases that aim to provide improved reference data for the IG loci[30,83].

By combining genetic variants with gene usage information across IGHV, IGHD and IGHJ genes derived from AIRR-seq data, we performed the first gene usage QTL analysis, assessing associations between 7297 common variants and 81 genes to identify polymorphisms explaining gene usage in the expressed IgM and IgG repertoire. These analyses revealed that half (52%) of common variants were associated with gene usage variation (based on statistical support after multiple-testing correction), impacting 59 (73%) genes in the IgM repertoire, with similar results in the IgG repertoire. This indicated that patterns in IgG are likely highly influenced by the gene usage composition initially established in IgM, as noted previously[20,21]. It is important to note that we chose to use a stringent $P$ value threshold (Bonferroni; $P < 9.2e{-}6$) to assess statistical support for the associations identified in this cohort. This should not be taken to mean that genes and variants not passing this threshold are biologically insignificant, but simply that larger sample sizes will be required to more fully characterize the

impact of IGH variants on the expressed repertoire. Further to this point, conditional analysis found that for 14 out of the 59 guQTL-associated genes in IgM, additional variance in gene usage could be explained by secondary polymorphisms, indicating that for at least a subset of IGH genes, interactions and additive effects across multiple variants will ultimately need to be resolved. However, it is critical that the collective effects of polymorphisms across the repertoire were clear when we compared repertoires between individuals based on genetic similarity. As expected[20,21], we found that usage patterns were more highly correlated in individuals sharing IGH genotypes. This indicated that overlapping signatures in the repertoires of different individuals may be possible to identify and characterize with greater resolution at the population level by simply taking into account IGH genetic data[22].

The guQTLs discovered here provide fundamental insights into the potential functional mechanisms underlying the development of the Ab repertoire in humans. First, the association between SVs and gene usage variation offer a straightforward model for how germline variants impact the repertoire. Specifically, our results indicated that SVs change the copy number of genes, directly modifying their usage frequency in an additive fashion, likely by influencing the probability that the SV-associated genes are selected by V(D)J recombination based on the number of chromosomes on which they are present. This pattern was observed for the majority of genes associated with SVs in our dataset and has been noted previously[40,42]. Interestingly, there were also genes

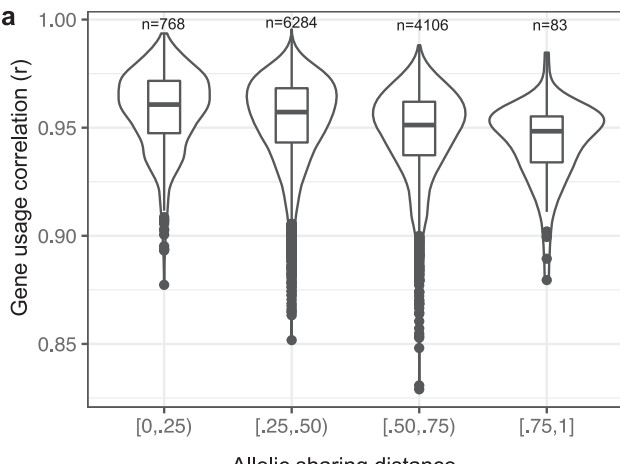

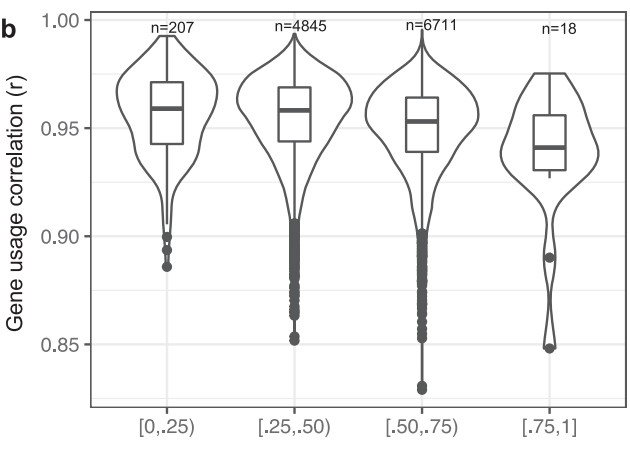

**Fig. 9 | Individuals sharing a greater number of guQTL genotypes have more correlated repertoire-wide IgM gene usage profiles. a, b** Pairwise intra-individual correlations (Pearson) of IgM usage for all genes, as well as allele sharing distance (ASD) for IGH SNV genotypes (lead guQTLs; all guQTLs) were calculated across individuals in the cohort. Violin plots show pairwise intra-individual repertoire-wide IgM gene usage correlations partitioned by ASD, calculated using either only lead guQTLs for all genes (**a**) or all guQTLs (**b**) for all genes (Bonferroni corrected). Boxplots display the median, 25th percentile, 75th percentile, and whiskers that extend up to 1.5 times the inter-quartile range (IQR) from the respective percentiles. Data points outside the whiskers are also plotted.

for which usage was impacted by neighboring SVs, even though the copy number of these genes was not directly altered, suggesting more complex mechanisms[42]. Beyond the effects of SVs, we found a significant number of SNVs associated with gene usage, all of which were in intergenic regions; again, this highlights the importance of our approach for capturing all IGH variant types, beyond just coding polymorphisms. Network analysis connecting genes with overlapping guQTL variants identified sets of genes whose usage patterns were coordinated. In many cases these genes were co-localized to specific regions of IGH, spanning 10s to 100s of Kbp. As with patterns observed for SVs, these signatures were illustrative of more complex regulatory mechanisms in the IGH locus. The regional effects observed appear consistent with studies of V(D)J recombination in model organisms. For example, the mouse IG loci partition into distinct regions, marked by specific regulatory marks, including TFBS and histone modification signatures, many of which, alongside RSS variation, have been associated with intra-gene V(D)J recombination frequency differences[32,84,85]. The mouse IG loci are also characterized by 3-dimensional structure, TADs and sub-TADs that are associated with complex interactions

between gene promoters and enhancers that coordinate V(D)J recombination in pre-B cells[35,53,86–88]. In contrast to mouse, functional genomic elements dictating V(D)J recombination in the human IGH locus have not been characterized in depth; nonetheless, our intersection of guQTLs with publicly available annotation sets revealed enrichments in cis-regulatory elements and TFBS involved in V(D)J recombination in animal models. This included CTCF and *EED* TFBS, as well as IGH regions marked by H3K4me3[54,61–63]. While fine mapping and functional validation of guQTLs is needed, this result provides initial evidence that the variants we identified likely influence the frequency at which IGH genes are selected during V(D)J recombination.

There is growing interest in developing predictive models for V(D)J recombination and repertoire diversity[89,90], and applying Ab repertoire profiling as a diagnostic tool for disease and clinical phenotypes of high public health relevance[91,92]. However, current models do not explicitly account for genetic factors, and the effects of this on model performance are not known[89,90]. Our results indicate that future work in this area should explore ways to integrate genetic data; this will likely be critical for better understanding commonalities and differences in repertoire signatures (e.g., public clonotypes[1,2]), ultimately leading to improved metrics for immune response monitoring and prediction modeling.

Here, we demonstrate that our data already provide an opportunity to more fully explore the potential roles of IGH polymorphism in Ab-mediated diseases. First, the direct overlap of GWAS SNVs and guQTLs indicate the potential for effects of GWAS variants to be mediated through genetic effects on Ab gene usage. Second, our results can directly inform our understanding of vaccine responsiveness, particularly as this pertains to efforts centered around the elicitation of targeted antibodies. Our analysis revealed that IGHV coding variation was in many cases linked to guQTLs, supporting previous reports indicating that usage patterns can coincide with amino acid differences[23,41,93], including those that are important for Ab–antigen interactions in infectious disease responses[23,41]. It is important to note that in many cases, allelic variants vary considerably between human populations[23,41], indicating that both population-level diversity and the role of germline variants in shaping the baseline B cell repertoire will need to be considered in interpreting vaccine response data[22,94].

While the dataset we have analyzed here represents the most comprehensive survey to date, it is likely that increasing the sample size will uncover additional genetic contributions to gene usage. Rarer and complex IGH variants will need to be better accounted for in future work, specifically those excluded from our analysis due to low frequency and genotyping coverage. In addition, as cohorts increase in size, additional insight will come from the consideration of other variables such as genetic ancestry, positive/negative selection, age, B cell subset and tissue[95–97]. Finally, the models utilized here could be extended to assess the contribution of IGH polymorphisms to other repertoire signatures, including N/P addition and CDR3 features, which also are influenced by heritable factors[20,21,38,90].

Collectively, our analyses provide a comprehensive picture of IGH polymorphism and Ab repertoire variation. These findings have the potential to reshape the way we conduct, analyze and interpret AIRR-seq data, and use these data to profile the Ab response in disease. As noted previously, the results provided here further illuminate the need for improving efforts to more fully explore the extent of IGH polymorphism in the human population, as a means to resolve the role of germline variation in Ab function and disease.

## Methods

### Ethics statement

This study complies with all relevant ethical regulations. The study and protocol were reviewed and approved by the Dana-Farber Cancer Institute (DFCI), Stanford University and University of Louisville Institutional Review Boards (IRBs). Informed consent for study participation

and collection of blood samples was obtained with research volunteers signing a consent form approved by the DFCI IRB.

## Long-read library preparation and sequencing

Genomic DNA was extracted from PBMC or PMN procured from Stanford University, Harvard University or STEMCELL Technologies (Vancouver, Canada); donor informed consent was obtained when necessary, following relevant ethical guidelines, and study protocols were approved by respective IRBs. Genomic DNA was processed using our published targeted long-read sequencing protocol[28]. Briefly, high molecular weight DNA (0.5–2 μg) was sheared using g-tubes (Covaris) and size selected using the 0.75% DF 3–10 Kbp Marker S1-Improved Recovery cassette definition on the Blue Pippin (Sage Science); library size ranges provided in Supplementary Fig. 1. The DNA was End Repaired and A-tailed using the standard KAPA library protocol (Roche). Barcodes were added to samples sequenced in multiplex pools and universal primers were ligated to all samples. PCR amplification was performed for 8–9 cycles using high-fidelity polymerase (LA-Taq or PrimeSTAR GXL, Takara) at an annealing temperature of 60 °C. Small fragments and excess reagents were removed using 0.7X AMPure PB beads (Pacific Biosciences). Libraries were hybridized to IGH-specific oligonucleotide probes (Roche; see reference[28]) and recovered using streptavidin beads (Life Technologies) prior to another round of PCR amplification for 16–18 cycles using either LA-Taq or PrimeSTAR GXL (Takara) at an annealing temperature of 60 °C.

Enriched IGH libraries were prepared for sequencing using the SMRTbell Express Template Preparation Kit 2.0 (Pacific Biosciences). DNA was treated with Damage Repair and End Repair mix to repair nicked DNA, followed by the addition of an A-tail and overhang ligation with SMRTbell adapters. These libraries were treated with a nuclease cocktail to remove unligated input material and cleaned with 0.45X AMPure PB beads (Pacific Biosciences). The resulting libraries were prepared for sequencing according to the manufacturer's protocol and sequenced as single libraries per SMRTcell with P6/C4 chemistry and 6 h movies on the RSII system, or as multiplexed libraries sequenced on the Sequel (3.0 chemistry; 20 h movies) or Sequel II/IIe system (2.0 chemistry; 30 h movies).

Generated targeted capture libraries had an average insert length of 6 Kbp, and were sequenced using the Pacific Bioscience (PacBio) RSII ($n = 40$), Sequel ($n = 40$), or Sequel IIe ($n = 74$) systems (Supplementary Table 1). This strategy confers two main advantages: (1) the sequencing polymerase passes over amplicons multiple times, allowing for the generation of highly accurate (high-fidelity, HiFi) reads (Supplementary Fig. 1a, b); and (2), for Sequel/IIe libraries, multiple samples are barcoded and sequenced in a single sequencing run. Critically, the high HiFi read quality overcomes historical concerns of error rates in long-read sequencing data (Supplementary Table 1), and error-correction steps performed during the assembly process increases the read base-level accuracy[98,99]. Previously, we have shown that assemblies produced from the older RSII platform have high base-level accuracy[28].

For a single sample, we prepared libraries for adaptive nanopore sequencing using the Ligation Sequencing Kit (Oxford Nanopore Technologies, ONT) and the NEBNext Companion Module for ONT Ligation Sequencing (New England Biolabs). 3 μg gDNA was used as input for these libraries. Entire purified libraries (5–50 fmol, per manufacturer's recommendation) were loaded onto R9.4.1 flow cells on the MinION Mk1C instrument (ONT). The experimental run was set up with no multiplexing, turning on enrich.fast5, and using human nanopore enrichment. Additionally, fast (or high accuracy) base calling was employed for a 72-h run. In addition to IGH, multiple genomic loci were targeted for sequencing in order to provide the minimum number of bases (17 Mb) required for adaptive sequencing. The IGH sequence targeted was from the custom reference used in this study (below).

## IgG and IgM antibody repertoire sequencing

For newly generated expressed Ab repertoire sequencing datasets, two distinct protocols were implemented for respective sets of samples. Total RNA was extracted from PBMCs using either the RNeasy Mini kit (Qiagen) or PureLink RNA Mini Kit (Ambion). AIRR-seq libraries were then generated using either a 5'RACE approach, or IGHV gene primer-based method. For IgG and IgM 5'RACE AIRR-seq, libraries were generated using the SMARTer Human BCR Profiling Kit (Takara Bio), following the manufacturer's instructions. Individually indexed IgG and IgM libraries were assessed using the Agilent 2100 Bioanalyzer High Sensitivity DNA Assay Kit (Agilent) and the Qubit 3.0 Fluorometer dsDNA High Sensitivity Assay Kit (Life Technologies). Libraries were pooled to 10 nM and sequenced on the Illumina MiSeq platform using the 300 bp paired-end reads with the 600-cycle MiSeq Reagent Kit v3 (Illumina). For the IGHV primer-based method, cDNA was first generated from 1 μg RNA using the Superscript RT III kit (Invitrogen) with Oligo-dT primer. AIRR-seq amplicons were generated from generated cDNA using a pool of IGHV primers (Supplementary Data 7) and one of two reverse primers targeting either IgM or IgG (Supplementary Data 7). Primers were pooled equimolar (0.1 μM/each), with 0.125 μl Taq polymerase (NEB) and 100 ng cDNA in 25 μl total volume. Cycling conditions were as follows: 94 °C denaturation for 3 min, 94 °C 1 min, 50 °C 1 min, 72 °C 1 min for 4 cycles, 94 °C 1 min, 55 °C 1 min, 72 °C 1 min for 4 cycles, 94 °C 1 min, 63 °C 1 min, 72 °C 1 min for 8 cycles, 72 °C 5 min, hold at 10 °C. Additional PCR cycles were conducted using a second set of primers with extension sequences, at a final concentration of 0.2 μM/each, with the following cycling conditions: 94 °C 1 min, 63 °C 1 min, 72 °C 1 min for 20 cycles, 72 °C for 5 min, hold at 10 °C. PCR amplicons were purified from 1% agarose gels (Zymo Research). Sequencing adapters and barcodes were added to purified PCR products using the KAPA HiFi HotStart kit and NEBNext 96 index kit, followed by and additional size selection and purification from 1% agarose gels (Zymo Research). Resultant barcoded libraries were quantified and pooled equimolar and sequenced on the Illumina MiSeq platform using the 300 bp paired-end reads with the 600-cycle MiSeq Reagent Kit v3 (Illumina).

Additional AIRR-seq datasets were downloaded from SRA for Nielsen et al.[18].

## Custom linear IGH reference

A custom linear reference for IGH was used that includes previously resolved insertion sequences[25] absent in GRCh38. This reference was previously used and vetted to generate high confidence variant call sets[28]. The reference was built from GRCh38 (chr14:105860 500–107043718). Partial sequences from GRCh38 were removed and additional insertion sequences were added from previously characterized SVs[25]. Specifically, sequence between chr14:106254581–106276923 (GRCh38) was swapped for a 10.8 Kbp duplication containing the *IGHV3-23D* gene from fosmids ABC9-43993300H10 and ABC9-43849600N9. Sequence between chr14:106317171-106363211 (GRCh38) and chr14:106403456–106424795 (GRCh38) was swapped for a 77.6 Kbp duplication haplotype containing IGHV genes *IGHV3-30, IGHV4-30-2, IGHV3-30-3, IGHV4-30-4, IGHV3-30-5, IGHV4-31* and *IGHV3-33* from fosmid clones ABC11-4715040OI4, ABC11-47354200D2 and ABC11-49598600E10; and a 75.8 Kbp insertion containing IGHV genes *IGHV3-38 IGHV4-38-2, IGHV3-43D, IGHV3-38-3, IGHV1-38-4* and *IGHV4-39* from fosmid clones ABC10-44084700I10, ABC10-44145400L1 and WI2-1707G1, respectively. A 37.7 Kbp complex SV with *IGHV3-9* and *IGHV1-8* genes derived from GRCh37 (chr14:106531320-106569343) was appended to the end of the reference separated by 5 Kbp of gap sequence ("N"). This reference sequence is available on github (https://github.com/oscarlr/IGenotyper).

## IGH locus assembly and variant detection

All targeted long-read datasets were processed using IGenotyper with default parameters[28]. IGenotyper uses BLASR[100], WhatsHap[101], MsPAC[102],

and Canu[98] to align reads, call and phase SNVs, phase reads, and assemble phase reads, respectively. Using the assemblies, IGenotyper uses the MsPAC multiple sequencing alignment and Hidden Markov model module to identify SNVs, indels and SVs. SVs not directly resolved were genotyped using HiFi read coverage and soft-clipped sequences in the assembly and in HiFi reads, and manually resolved using BLAST and custom python scripts. SVs that could not be resolved using HiFi reads or assemblies were not genotyped and were not included in downstream analyses. All SV genotypes were visually inspected using Integrated Genome Viewer (IGV) screenshots generated from an IGV batch script.

### Characterizing novel alleles and expanding the IGH allele database

Novel alleles for IGHV, IGHD and IGHJ genes supported by 10 HiFi reads (exact matches) or found in 2 or more individuals were extracted from the assemblies of each sample. Novel alleles were defined as those not found in the IMGT database (release 202130-2). Allele sequences that aligned to IMGT alleles with 100% identity were also characterized as novel, if the putative novel allele was annotated from a gene in the assembly that was different from the gene assignment in the IMGT database. The non-redundant set of novel alleles was appended to the IMGT database for IgM/IgG repertoire sequencing analyses conducted in this study. A BLAST database was created using makeblastdb version 2.11.0+. Gapped sequences for the novel alleles were generated using the IMGT/V-QUEST server[103].

### Processing AIRR-sequencing data

Paired-end sequences ("R1" and "R2") were processed using the pRE-STO toolkit[104]. All R1 and R2 reads were trimmed to $Q = 20$, and reads <125 bp were excluded using the functions "FilterSeq.py trimqual" and "FilterSeq length," respectively. Constant region (IgM and IgG) primers were identified with an error rate of 0.2 and corresponding isotypes were recorded in the fastq headers using "MaskPrimers align."

For sequencing datasets without unique molecular identifiers (UMIs), R1 and R2 reads were assembled using "AssemblePairs align" and resulting merged sequences <400 bp were removed using "FilterSeq length." Identical sequences were collapsed and read duplicate counts ("Dupcounts") were recorded. For sequencing datasets with UMIs, the 12 base UMI, located directly after the constant region primer, was extracted using "MaskPrimers extract." Sequences assigned to identical UMIs were grouped and aligned using "ClusterSets" and "AlignSets muscle," and then consensus sequences were generated for each unique UMI set using "BuildConsensus." Identical sequences with different UMIs were collapsed and read duplicate counts ("Dupcounts") were recorded. Collapsed consensus sequences represented by <2 reads were discarded.

Processed AIRR-seq fastq files were split by isotype using the "SplitSeq.py group" function from Immcantation[104]. Samples with <100 reads per isotype were removed. Following the application of this filter, the mean number of merged consensus sequences per repertoire ranged from 465 to 109,250 (mean=26,036), with lengths ranging from 318 to 510 bp. Fastq files were aligned to the expanded database, including IMGT and novel alleles identified in our cohort, using "AssignGenes.py igblast" to generate Change-O[105,106] files. Productive reads were specifically selected using the "ParseDb.py split" command. Assignments to genes found to be deleted from both chromosomes in genomic datasets for a given sample were removed from the Change-O. Reads assigned to multiple alleles were re-assigned to a single allele if the genomic data revealed that only one of the alleles was present. Clones were detected using the modified Change-Os with the "shazam distToNearest" command and "model=ham," normalize="len" parameters, "shazam findThreshold" (parameters: method="gmm," model="gamma-gamma"), and "DefineClones.py (parameters: –act set –model ham –norm len –mode allele)" commands. IgM and IgG

repertoires with fewer than 200 clones identified were excluded from downstream analysis.

### Calculating gene usage among defined clones

A $m \times n$ clone count matrix **C** was created, where $m$ are the genes and $n$ are the samples. Each value in **C** represented the number of clones counted for a given gene in a given sample. Due to sequence similarity, duplicated genes were summed into a single entity. The counts of the following genes were combined:

1. *IGHV3-23* and *IGHV3−23D*
2. *IGHV3-30*, *IGHV3-30-3*, *IGHV3-30-5*, and *IGHV3-33*
3. *IGHV1-69* and *IGHV1-69D*
4. *IGHD4-4 and IGHD4-11*

*C* was batch corrected (3 batches) using ComBat-seq[107] to produce an adjusted count matrix *C'* to account for differences between the three AIRR-seq datasets used. The fractions of clones per gene or gene set ($m$) was calculated from *C'* across each sample ($n$).

The following set of F/ORF genes were removed or not analyzed:

1. *IGHD5-5*: In all cases where *IGHD5-5* was identified through IgBLAST, the AIRR-seq reads were assigned to *IGHD5-5*01* and *IGHD5-18*01*, or *IGHD5-5*01*, *IGH5-18*01* and additional alleles. The genes *IGHD5-5* and *IGHD5-18* were not combined because there were AIRR-seq reads aligned solely to *IGHD5-18*.
2. *IGHV3-16*: No AIRR-seq reads were assigned to *IGHV3-16*.

### Selecting common variants for gene usage QTL analysis

SNVs with a HWE value less than 0.000001 were filtered using bcftools[108]. SNVs found in less than 5 individuals were removed if they did not have HiFi read support. The SNVs passing these stringent quality control thresholds were used to impute missing genotypes using Beagle[109] (v228Jun21.220). The resulting SNVs were again filtered if they contained a HWE value less 0.000001. Common SNVs were selected if they were genotyped in at least 40 individuals and had a MAF equal to or greater than 0.05. The same criteria were applied to SNVs selected for conditional analysis.

Indels and SVs, excluding large SVs (>9 Kbp), were split into two categories based on whether they overlapped tandem repeat regions. Tandem repeat regions on the custom reference were determined using Tandem Repeats Finder[110] with parameters (match = 2, mismatch = 7, delta = 7, PM = 80, PI = 10, Minscore = 10, MaxPeriod = 2000). Events overlapping tandem repeats were genotyped again in all the samples using the dynamic programming algorithm from PacMonSTR[111]. Events were merged using a custom python script (https://github.com/oscarlr-TRs/PacMonSTR-merge). Tandem repeat events with an alignment score between the motif and the copies in the assemblies lower than 0.9 were removed. Tandem repeat alleles were defined by a difference of a single motif copy. Tandem repeat events with an allele occurring at a frequency greater than 0.05 was considered common. An expansion or contraction greater than 50 bps relative to the reference was considered a tandem repeat SV. Indels and SVs from IGenotyper outside of tandem repeats across all samples were merged. Manual inspection showed high concordance between event sizes and sequence content. In cases where a discordance was observed between event sizes, the max size was selected. Samples were genotyped as homozygous reference for indels and SVs if no event was detected and both haplotypes were assembled over the event. Indels and SVs with a MAF greater than 0.05 were selected.

All SVs were genotyped using IGenotyper and manually inspected using IGV. SVs with a MAF less than 0.05 were not included in the guQTL analysis (Supplementary Data 2).

### Gene usage QTL analysis

Genotypes at SNVs, complex SVs and mSVs were tested for association with usage using ANOVA and linear regression. Association tests for all

other variant types, indels, non-complex SVs and large SVs (excluding mSVs) were conducted using linear regression. Both models included age and AIRR-seq sequencing platform as covariates ($n = 3$). A linear regression was used to extract additional metrics (e.g., beta coefficients and $R^2$ values). Associations were corrected for multiple testing using Bonferroni on a per-gene level. Variants with an LD of 1 ($r^2$) were treated as a single variant during correction, representing only a single association test. Conditional analysis was performed in the same manner using all variant types with the same filters applied to the initial call sets.

### Network analysis of variants associated with multiple genes

Variant and gene pairs for variants significantly associated with more than 1 gene in the IgM repertoire were selected. A graph using the networkx python library (networkx.org) was created with genes as nodes and edges connecting genes/nodes if the same variant was associated with both genes. An edge weight was given for each time nodes were connected. The graph was pruned such that the edge weights were greater than 2. Cliques were identified using the find_cliques function.

### Regulatory analysis

ENCODE cCREs were downloaded from the UCSC Genome Browser under group "Regulation," track "ENCODE cCREs," and table "encode CccreCombined." ENCODE transcription factor binding site data were also downloaded from the UCSC Genome Browser under group "Regulation," track "TF Clusters," and table "encRegTfbsClustered." SNVs associated with gene usage were overlapped with both tracks and an enrichment in both tracks over all SNVs overlapping each track was calculated using a one-sided Fisher Exact Test.

### GWAS analysis

Variants identified by GWAS with an association $P$ value lower than 4e−6 were downloaded from the NHGRI-EBI GWAS catalog (https://www.ebi.ac.uk/gwas/api/search/downloads/full). Significant variants from this study were intersected with GWAS variants.

### Reporting summary

Further information on research design is available in the Nature Portfolio Reporting Summary linked to this article.

## Data availability

The IGH locus long-read sequencing data and AIRR-seq datasets generated in this study have been deposited in the BioProject repository PRJNA555323, under accession numbers SRX19355477–SRX19354801 (IGH locus) and SRX19355879–SRX19536764 (AIRR-seq). Previously published AIRR-seq datasets are available in the Sequence Read Archive (SRA) under accession numbers SRS3786791–SRS3786902. Metadata and summary statistics for this study are provided in Supplementary Data 1–7.

## Code availability

Code used to resolve additional SVs can be found on GitHub: https://github.com/oscarlr/bioinformatics#merging-contigs[112]. Tandem repeat genotyping and processing code, PacMonSTR and PacMonTSTR-merge, can be found here: https://github.com/oscarlr-TRs/PacMonSTR[113], https://github.com/oscarlr-TRs/PacMonSTR-merge[114].

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

## Acknowledgements

We grateful for constructive feedback provided by three anonymous reviewers. O.L.R., C.A.S., K.S., W.S.G., J.T.K., M.L.S., and C.T.W. were supported in part by grants R24AI138963 and R21AI142590 from the National Institute of Allergy and Infectious Diseases. O.L.R. was also supported by the Zuckerman postdoctoral fellowship. W.A.M. and H.K. were supported in part by R01AI121285 and 5U01AI165442 from the National Institute of Allergy and Infectious Diseases. Y.S. was supported in part by a postdoctoral intersect fellowship from the American Association of Immunologists. S.D.B. is supported in part by National Institutes of Health grants R01AI127877, R01AI130398, R01AI125567, U19AI057229, U19AI104209, and U19AI167903.

## Author contributions

O.L.R., M.L.S., W.A.M., and C.T.W. conceived and planned the study. O.L.R., Y.S., and D.T. performed computational experiments. C.A.S., K.S., W.S.G., J.T.K., and H.K. performed wet lab experiments. W.A.M., M.L.S., and C.T.W. supervised the study. H.K., K.J.L.J., S.D.B., W.A.M., and C.T.W. provided samples and data. All authors read and approved the final manuscript.

## Competing interests

The authors declare no competing interests.
