## [Peer Review File · Nature Communications]

Genetic variation in the immunoglobulin heavy chain locus shapes the human antibody repertoireREVIEWER COMMENTS

Reviewer #1 (Remarks to the Author):

Review of NCOMMS-22-35083-T, Rodriguez et al, "Genetic variation in the immunoglobulin heavy chain locus shapes the human antibody repertoire", 2022

Thanks for the opportunity to review this interesting paper. The authors study germline genetic variation at the IGH locus (which encodes the Immunoglobulin heavy chain, i.e. antibodies) and its impact on antibody responses in humans. This is an important topic, because the antibody response is one of the key determinants of the outcome of infections as well as immune-mediated conditions, and because current genetic variation catalogues do not access this and similar complex regions well (mainly due to being based inference from short-read sequence and alignment to a single reference genome). To solve this the authors compile a dataset of long-read sequencing data, including some newly generated sequences, using DNA from 154 healthy adults. They use this to call germline genetic variant including structural variants. In the same adults they sequence the immune receptor repertoire using a method called AIRR-seq. This setup then allows them to test for association between genetic variants and antibody usage. This work is a continuation of earlier work by the authors to develop methods and datasets informing on this topic, but goes further in applying this to a larger sample and by linking to antibody variation.

In general I have a very positive view of this paper: it is well-written, addresses an important topic, has appropriate methods and provides a substantial update on current knowledge. This includes some surprising findings that will be of considerable interest, such as the observation that germline variation explains a large proportion of some antibody responses (as against the view that somatic recombination/hypermutation in T/B cells is the key generators of antibody diversity and hence infection response), and the observation that functional mutations appear clustered in sites of V(D)J recombination. Additional insights include addition of previously unobserved IGHV/D alleles to the imgt.org database. I have no major concerns but a few minor comments are given below.

Minor comments:

1.1 Main text reads well but in some places I felt it would benefit from slightly more methodological detail being added (even though in some cases this is well covered in Methods). It is not indicated how AIRR-seq works, and stating this would be helpful because important for interpretation of the

association results. (At the very least AIRR-seq should be referenced on first use, e.g. lines 99 and/or 117, but I would prefer wording to be added to give an overall sense of how that antibody profiling is done.). For AIRR_seq this detail is not given in methods either. Are particular cell types needed, how are they captured, how is sequencing performed etc.? (Reference 18 is not the 'AIRR-seq reference I don't think, meanwhile searches for 'AIRR-seq' lead to information on the AIRR community but not obviously a methods publication, so including this information here would be very helpful to readers not familiar with the method.)

1.2 Another example on lines 643 onwards: Little information is given in main text about how the SV genotyping is performed, though this is explained in Methods and uses the authors' previously published IGenotyper method. This is important because they are both identifying and genotyping SVs and this could be quite challenging in principle - they have a strong method for this. Lines 136 onwards talk about 'genotyping' results but they are both discovering and genotyping SVs in this sample. This does not even refer to the method (IGenotyper), nor does it give any hints as to how this is done. This is important because identifying and calling SVs is quite challenging problem in general, and I would prefer the authors to convince me they hve done something robust and correct here. Looking at the Methods, line 643-644 says only "IGenotyper uses the MsPAC multiple sequencing 644 alignment and Hidden Markov model module to identify SNVs, indels and SVs", while the previous sentence talks about read alignment SNV calling and phasing, and assembly. My suggestion: add lines before line 136 to state briefly that the results are based on the author's previously published method, which uses an initial round of phased SNV calls made against a custom reference sequence to generate haplotype-resolved assemblies (independently for each sample). These are then combined in a multiple sequence alignment and SVs are inferred using MsPAC, which models the MSA using an HMM. (If I've got this right).

1.3 Also were the haplotype-resolved assemblies validated in any way? (Relevant because assemblers such as Canu can produce incorrect assemblies, which might confound SV calling in principle.) If so I suggest stating that here, if not I suggest stating that as well i.e. acknowledging that errors in haplotype-resolved assemblies could still occur using this method but we should trust the SV calls because <reasons>... (e.g. I think lines 646-647 are relevant.). In short, can you tell us how you are confident the discovered SVs are true SVs and not assembly artifacts and/or be clear if there is any uncertainty in this.

1.4 (I'm aware that much of the above is strictly methods, and it relies on a previously published approach, but I think it is worth including at least a few additional pointers on this in main text, as it gives the reader confidence in the SVs and genotypes that are analysed in the rest of the paper. Some additional detail in Methods would also be helpful I think.)

1.5 On a related note, I couldn't easily see information about what input material is used for the above methods. In particular are specific cell types needed to generate DNA for the assemblies? (This is important because there region somatically recombines in B cells; if the underlying reads are from whole blood then a proportion will be from already-recombined cells. How are these reads dealt with?).

Some indication of how this is dealt with e.g. are B cells excluded from the input material, or if not is this dealt with bioinformatically, would be helpful. This is particularly true because the data arise from multiple experiments.

2.1 Lines 206-280 onwards describe the association analysis between IGH germline genetic variants and gene usage in the antibody expressed repertoire. This is well conducted and the authors have provided full summary table which is appreciated. However I felt the section itself could be written in a way that was easier to understand - a couple of comments on this below.

2.2. This section contains lots of numbers of variants, including listing lots of #s of variants associated with IgM or IgG repertoires at the chosen P-value threshold. However it must be noted that the chosen P-value threshold (based on Bonferroni) is a pragmatic rather than especially meaningful threshold. Hence the 'numbers of significant associations' don't in themselves have much biological meaning and although I understand why the authors have done this (it is a defensible approach that lets them state concrete numbers) the emphasis on them here seems a bit misplaced. For example, a major result of this section is that (even with the relatively conservative Bonferroni threshold and smallish sample size) ~80% of IGHV and IGHD genes appear to be associated with germline genetic variants - that is pretty interesting and I think is a key result of the paragraph starting on line 208. The actual numbers of variants associated with these genes seem less relevant, but for this reader they distracted me from understanding this point properly.

2.3. (What happens if you reduce the Bonferroni threshold - do you get many more genes with associations?)

2.4. Similarly this paragraph ends with a description of the supplementary table (lines 226-231), how about putting that into a Supplementary Table legend and simply writing that comprehensive summary statistics for these association tests is provided.

2.5. Could the Supplementary Table contain standard errors as well as effect size estimates please? (These can be inferred from the P-value but having them stated would be better.)

2.6. A suggestion would be: combine paragraphs on line 208 and 283 into a single paragraph that says briefly how association analysis & conditional analysis was conducted. Then summarise the main findings in subsequent paragraphs, possibly with specific subheadings e.g. for non-SV-driven associations, as needed.

2.7. I should add that my comments 2.1-2.6 are largely stylistic i.e. up to authors preference. In general the material in this section is interesting and does get across the key points and has defensible methods.

2.8 Another comment I had on this section was that the authors do not consider possible confounding. This could be important because the dataset is quite diverse e.g. individuals of varying ethnicities and data generated in different experiments on different platforms and so some examination of whether there is evidence that the associations correlate with any of the other experimental covariates seems needed. In saying this I note that this is an observational study; as such it can't decide what the underlying causal pathway of any association actually is, and there is a strong a priori case that genetic variation might be the cause of gene usage variation. Hence, my suggestion would be to include a covariate analysis as supplementary information and, for any associations where a covariate seems to 'explain' or partly explain the association this should be noted as a possible alternate pathway. (Specifically I am not suggesting to alter the currently presented association tests to include covariates, and I don't think this would make sense to do.)

3.1. My other comment is that the Discussion (lines 433-572) is very long! (5 1/2 pages double-spaced.). I suggest the authors cut this down especially where this simply duplicates results. I would say the author's results are strong enough that they don't need to spend time re-justifying their efforts or repeating results. E.g. I'm not sure the paragraph on line 447 is needed, lines 456-468 seem to repeat results, so do lines 475-487, etc..

4.1 Re: figures. The authors have gone for a 'every figure is a massive multi-panel' figure type of approach. There is lots of information in figures but I personally think that removing some of this to supplementary figures, adding clarifying wording / labels and simplifying would help the presentation. Concrete examples below:

4.2. For example, in Figure 1, are panels b-d and f really needed? (These SVs are shown in panel a. and the only additional info is length, so they are repeating information. How about printing the lengths in panel a). Is the SV on panel f also in panel a? I couldn't spot it or these IGHD genes - if not why not? Adding more space between panels would also help.

4.3. Many of the panels could be improved with better labelling (including a title). For example, the two CDFs in panel h are labelled 'individuals' on y axis - they mean proportion of individuals, presumably. (I had to work hard to understand what these panels meant. I think it is: 'for each number of deleted gene alleles up to 50, this is the proportion of individuals in that population with that number or fewer deleted gene alleles'?)

4.4. In Fig 1 panel j it's not immediately obvious what 'SNVs' and 'Common SNVs' are and the bracket seems to say some of the common SNVs are rare (reading the legend I see this is about what dbSNP says are rare, but that's nonobvious in the figure); panel l is just showing three numbers (I think?) so doesn't really need to be there at all.

4.5. Fig 2 panel a confused me too. The first four tracks are visually similar and reflect somewhat related information so it's not immediately obvious what to look at first. Then, they use colours that mean different things in each track but have overlapping colour schemes. It's not stated what the P-value used is (is it the smallest P-value across variants for the gene). I assume the 'usage fold change' is an effect size derived from the regression estimate, but I was expecting to see this presented as a point and 95% confidence interval instead of a bar. The 'variant type' also doesn't need to be bars at all (the height is irrelevant) and the use of overlapping colours is again confusing.

4.6. Fig 2 panel e looks cool but (in common with several other panels) there's no indication of what it is about without reading the legend. (In this case, the legend for panel e refers to panel f as well which isn't there (at least in the combined pdf I'm using for this review)). Similarly the Fig 3 panels don't give any indication what they relate to without reading the legend.

4.7. In short - I think the figures would benefit from an editorial review of the figures to substantially simplify and clarify. (Similar to many of my comments this is obviously stylistic rather than about the results themselves, which seem strong.)

5.1 Line 758 "The datasets generated during and/or analysed during the current study are available from the corresponding author on reasonable request." -> please can the authors deposit the data, ideally including the haplotype-resolved assemblies and AIRR-seq results, in an open repository and include details here?

Reviewer #2 (Remarks to the Author):

Comments on the manuscript "Genetic variation in the immunoglobulin heavy chain locus shapes the human antibody repertoire" by O.L. Rodriguez et al. (2022) submitted to Nature Communications

Rodriguez et al. in their manuscript "Genetic variation in the immunoglobulin heavy chain locus shapes the human antibody repertoire" show that a person's antibody repertoire is strongly affected by genetic variation in the IGH locus. It is an impressive study, which combines germline IGH long-read sequencing

data and repertoire (AIRR-seq) data from the same individuals. The studied cohort includes individuals of various ethnicities, which gives this study another dimension. The results presented in this study are of great importance to the field and provide new insights into how antibody repertoires are generated.

I do not have experience with long-read sequencing, only with AIRR-seq; however, the methods and analytical approaches sound reasonable to me.

The conclusions of the study are supported by the data presented.

Although the code for the entire analysis is not made publicly available (apart from one custom python script), the authors provide some details about the packages and functions used, as well as modifications to the default parameters. Perhaps a bit more detail could be included in the methods sections “Gene usage QTL analysis”; “Network analysis of variants associated with multiple genes” and “Regulatory analysis”.

Some minor edits might improve clarity and readability:

Samples used in the study: In the Materials and Methods section, the authors say that PBMCs or PMN were obtained from either Stanford University, Harvard University or STEMCELL Technologies. Could you add the information about the cell source and the institution from which the samples were obtained to the Supplementary Table 1?

Clone definition: How were the clones defined in the AIRR-seq analysis? Can the authors make it more clear in the results section how they defined a clone? The criteria for defining a clone can vary between different studies.

Figures: The figures are sometimes too complex and/or difficult to read. Please, reconsider if all parts of all figures are absolutely necessary, or perhaps if it is possible to show certain data differently:

- Figure 1; mainly a) and h): The color scheme the authors use in Fig. 1a) and h) makes it very difficult to distinguish the data shown. Different shades of orange are difficult to distinguish both on screen and when printed; the same goes for the different shades of teal.

- Figure 1 j) and l): In Fig. 1j) the authors use the color red for “No”, but in Fig. 1l) they use the same color for “Yes”. For the sake of consistency and to make it easier for the readers, it might be good to keep the same colors for “Yes” and “No” in the same figure. However, I am not sure if these bar plots

are absolutely necessary to be presented in this way. Would it be perhaps better to have these as tables rather than bar plots, so that it is easier to see the exact values?

- Figure 2 – part f) is missing: I am not sure if it is due to the file merging, but I cannot see part f in Figure 2, despite it being mentioned in the figure legend. I can only see a)-e).

- Figure 3d): This plot is not easy to read. In the results section (line 320) it says: “We found 2607 (66%) guQTL variants associated with > 1 gene (Fig. 3d)”; however, this information is not easy to see from figure 3d. Could the actual values be perhaps put on top of the bars in the barplot? The x-axis labels are also a little bit confusing. Having the x-axis labels as 1, 2, 3,etc. would be easier to read than 2.5, 5.0, 7.5, etc. More labels on the y-axis might also be helpful.

- Figure 3e) and f): Would it be possible to make these plots bigger and put them into a separate figure? With their current size, the text is too small to read. Also, with the overlapping text, the genes names (especially those in the middle) are hard to see.

Discussion

Line 538: In what way does the reference no. 91 (Ohlin (2020) Front. Immunol.) support your statement? I had a look at the referenced paper, but I am unsure how it relates to the point being discussed in lines 536-539.

Data Availability: What is the reason for not making the data publicly available in a repository?

Supplementary data:

- Would it be possible to label the figures or at least number the pages, so that it is easier to distinguish which figure is which?

- Could the figure legends appear under the figures, at least for those figures that do not take up the entire page?

- Figure 11: the figure itself is not very clear, and the figure legend is not informative either.

- Figure 13: Same as in Fig.3 in the main text; the gene names and the numbers are very difficult to see. Especially the text that lies directly on top of those dots is almost impossible to see.

Apologies for the delay in reviewing this manuscript and wishing the authors happy holidays!

Reviewer #3 (Remarks to the Author):

the study reported here is of immense importance to immunology and the study of multivariate gene areas. The authors have applied cutting edge sequencing methods and sequence analysis methods to characterize the heavy chain germline locus in its entirety in 154 individuals. The novel data set by itself should have a great impact on the field as it is nearly unique in describing this locus. This dataset radically increases our knowledge of the true makeup of germline diversity. Over and above these findings the authors also show (through sequencing and analyzing of somatic repertoires from these individuals) that there is a relationship between the genotype (And number of copies in the germline) of a different gene types and their somatic expression. Moreover, we can see from this study how frequent some but not all SV and SNV are we can now start to characterize what part of diversity in germline populations influences somatic germline gene usage and expression.

There are no major issues with the paper but i have a few minor ones that should be addressed before publication:

- 1) A glossary of acronyms should be made of all acronyms from SV and SNV to QTL and its types.
- 2) the use of p-values to describe effect size. This is a bad and misleading practice that should not remain in the paper.

thus for instance in figure 5 genes should be ranked by the skew in their expression not by their p value. The description in the figure caption also is misleading. There is no such thing as "3 most significant". As the authors themselves state all genes colored in pink are significant with significance set by the p value threshold - $p < 0.05$. thus they are all "equally" significant. The exact p values could be influenced by effect size in terms of size of skew but also by other things.

other figures suffering from this issue are

figure 2a and figure 3, b,c and g and supplemental figure 7

3) the network figures and especially in the sup. materials are indecipherable. it is very hard to read the gene names and numbers describing the strengths of edge connection. could the author redraw the networks and especially figure 12 where one can not read the gene names.... they do not need to make this figure a printable page size. make it as large as needed so we can see all the gene names.

4) the figure captions in sup materials are lacking. please describe these figures with as much detail as the figures in the paper.

REVIEWER COMMENTS

Reviewer #1 (Remarks to the Author):	1
Reviewer #2 (Remarks to the Author):	7
Reviewer #3 (Remarks to the Author):	10

We are grateful for the time given to our manuscript and the thoughtful comments and suggestions made by the reviewers. Please find our responses to each of the comments below, addressed in turn (“underlined text”).

REVIEWER COMMENTS

Reviewer #1 (Remarks to the Author):

Review of NCOMMS-22-35083-T, Rodriguez et al, "Genetic variation in the immunoglobulin heavy chain locus shapes the human antibody repertoire", 2022

Thanks for the opportunity to review this interesting paper. The authors study germline genetic variation at the IGH locus (which encodes the Immunoglobulin heavy chain, i.e. antibodies) and its impact on antibody responses in humans. This is an important topic, because the antibody response is one of the key determinants of the outcome of infections as well as immune-mediated conditions, and because current genetic variation catalogues do not access this and similar complex regions well (mainly due to being based inference from short-read sequence and alignment to a single reference genome). To solve this the authors compile a dataset of long-read sequencing data, including some newly generated sequences, using DNA from 154 healthy adults. They use this to call germline genetic variant including structural variants. In the same adults they sequence the immune receptor repertoire using a method called AIRR-seq. This setup then allows them to test for association between genetic variants and antibody usage. This work is a continuation of earlier work by the authors to develop methods and datasets informing on this topic, but goes further in applying this to a larger sample and by linking to antibody variation.

In general I have a very positive view of this paper: it is well-written, addresses an important topic, has appropriate methods and provides a substantial update on current knowledge. This includes some surprising findings that will be of considerable interest, such as the observation that germline variation explains a large proportion of some antibody responses (as against the view that somatic recombination/hypermutation in T/B cells is the key generators of antibody diversity and hence infection response), and the observation that functional mutations appear clustered in sites of V(D)J recombination. Additional insights include addition of previously unobserved IGHV/D alleles to the imgt.org database. I have no major concerns but a few minor comments are given below.

Minor comments:

1.1 Main text reads well but in some places I felt it would benefit from slightly more methodological detail being added (even though in some cases this is well covered in Methods). It is not indicated how AIRR-seq works, and stating this would be helpful because important for interpretation of the association results. (At the very least AIRR-seq should be referenced on first use, e.g. lines 99 and/or 117, but I would prefer wording to be added to give an overall sense of how that antibody profiling is done.). For AIRR_seq this detail is not given in methods either. Are particular cell types needed, how are they captured, how is sequencing performed etc.? (Reference 18 is not the 'AIRR-seq' reference I don't think, meanwhile searches for 'AIRR-seq' lead to information on the AIRR community but not obviously a methods publication, so including this information here would be very helpful to readers not familiar with the method.)

Thank you for this suggestion. We have added the following text (lines 134-140):

AIRR-seq is a powerful technique for analyzing the diversity and composition of expressed adaptive immune receptors. Within a given B cell during development, a single IGHV, IGHD and IGHJ gene are somatically rearranged at the genome level. These recombined IGHV, IGHD, and IGHJ segments are transcribed and spliced together with a constant (IGHC) gene, which determines the receptor isotype (e.g., IgM or IgG). AIRR-seq molecular protocols allow for the selective sequencing of VDJ receptors through the amplification of cDNA (or rearranged genomic DNA) using primers targeting specific IGHV, IGHJ and/or IGHV genes.

1.2 Another example on lines 643 onwards: Little information is given in main text about how the SV genotyping is performed, though this is explained in Methods and uses the authors' previously published IGenotyper method. This is important because they are both identifying and genotyping SVs and this could be quite challenging in principle - they have a strong method for this. Lines 136 onwards talk about 'genotyping' results but they are both discovering and genotyping SVs in this sample. This does not even refer to the method (IGenotyper), nor does it give any hints as to how this is done. This is important because identifying and calling SVs is quite a challenging problem in general, and I would prefer the authors to convince me they have done something robust and correct here. Looking at the Methods, line 643-644 says only "IGenotyper uses the MsPAC multiple sequencing 644 alignment and Hidden Markov model module to identify SNVs, indels and SVs", while the previous sentence talks about read alignment SNV calling and phasing, and assembly. My suggestion: add lines before line 136 to state briefly that the results are based on the author's previously published method, which uses an initial round of phased SNV calls made against a custom reference sequence to generate haplotype-resolved assemblies (independently for each sample). These are then combined in a multiple sequence alignment and SVs are inferred using MsPAC, which models the MSA using an HMM. (If I've got this right).

We extended our description of SV detection and genotyping in the main results to include (lines 178-181):

We utilized our previously published tool, IGenotyper, to generate haplotype-resolved assemblies. We then used these contigs in conjunction with haplotype-specific HiFi reads to create a manually curated genotype call set for large SVs (> 9 Kbp; Fig. 1b) within 8 regions of IGH, excluding genotypes in samples that were not supported by haplotype-specific HiFi reads.

1.3 Also were the haplotype-resolved assemblies validated in any way? (Relevant because assemblers such as Canu can produce incorrect assemblies, which might confound SV calling in principle.) If so I suggest stating that here, if not I suggest stating that as well i.e. acknowledging that errors in haplotype-resolved assemblies could still occur using this method but we should trust the SV calls because <reasons>... (e.g. I

think lines 646-647 are relevant.). In short, can you tell us how you are confident the discovered SVs are true SVs and not assembly artifacts and/or be clear if there is any uncertainty in this.

We added text detailing that the SVs were validated by directly evaluating HiFi read coverage support in IGV. See response to question 1.2.

1.4 (I'm aware that much of the above is strictly methods, and it relies on a previously published approach, but I think it is worth including at least a few additional pointers on this in main text, as it gives the reader confidence in the SVs and genotypes that are analysed in the rest of the paper. Some additional detail in Methods would also be helpful I think.)

We added text detailing that the SVs were validated by evaluating their read coverage profile in IGV. See response to question 1.2.

1.5 On a related note, I couldn't easily see information about what input material is used for the above methods. In particular are specific cell types needed to generate DNA for the assemblies? (This is important because there region somatically recombines in B cells; if the underlying reads are from whole blood then a proportion will be from already-recombined cells. How are these reads dealt with?). Some indication of how this is dealt with e.g. are B cells excluded from the input material, or if not is this dealt with bioinformatically, would be helpful. This is particularly true because the data arise from multiple experiments.

Thank you for noting this, it is indeed important to note these points in the text. The results section "Paired IGH targeted long-read and antibody repertoire sequencing" describes that the DNA used for PacBio sequencing and assembly was from PBMCs and PMNs. We added additional text in this section to describe that the DNA that could come from somatically rearranged chromosomes within B cells is very small and that no recombined assemblies were observed. We would not expect these recombined DNA fragments to drastically impact our assemblies because (1) PBMCs typically contain a small proportion of B cells, and (2) a population of B cells contains a large diversity of V(D)Js from different regions of the locus; thus in any one region, we would expect the native DNA fragments captured from non-B cells to far outnumber that contributed by individual rearranged chromosomes. Visual inspection of mapped capture reads to our reconstructed assemblies supports this conclusion. Furthermore, we have extensive experience working with data in B cell derived LCLs, in which dominant V(D)J rearrangements can be observed – such signatures are evident in these data, and easily discernible (see references PMID: 34898642 and PMID: 33072076).

We have added this text the manuscript:

PBMCs are composed of 70-90% lymphocytes, with B cells making up only 5-10% of the total number of lymphocytes. As a result, we would not expect DNA derived from individual B cell lineages to make significant contributions to the resolved assemblies.

In contrast to observations made using lymphoblastoid cell lines^{28,45}, no V(D)J rearrangements were observed in the assemblies, demonstrating that sequencing reads from recombined B cell-derived DNA did not contribute to the assembly process.

2.1 Lines 206-280 onwards describe the association analysis between IGH germline genetic variants and gene usage in the antibody expressed repertoire. This is well conducted and the authors have provided full summary table which is appreciated. However I felt the section itself could be written in a way that was easier to understand - a couple of comments on this below.

[Responses below]

2.2. This section contains lots of numbers of variants, including listing lots of #s of variants associated with IgM or IgG repertoires at the chosen P-value threshold. However it must be noted that the chosen P-value threshold (based on Bonferroni) is a pragmatic rather than especially meaningful threshold. Hence the 'numbers of significant associations' don't in themselves have much biological meaning and although I understand why the authors have done this (it is a defensible approach that lets them state concrete numbers) the emphasis on them here seems a bit misplaced. For example, a major result of this section is that (even with the relatively conservative Bonferroni threshold and smallish sample size) ~80% of IGHV and IGHD genes appear to be associated with germline genetic variants - that is pretty interesting and I think is a key result of the paragraph starting on line 208. The actual numbers of variants associated with these genes seem less relevant, but for this reader they distracted me from understanding this point properly.

Thank you for this suggestion. We agree with the reviewer that the key point is the number of genes with statistical support, and that this point should be clear. We have removed the majority of the text and added a table (Table 2) with the counts of variants and genes to hopefully address this.

2.3. (What happens if you reduce the Bonferroni threshold - do you get many more genes with associations?)

Yes! In the supplementary table we have included the p-value, so the number of genes with significant hits can be further evaluated by reducing the threshold. We have performed this analysis and mentioned it in the discussion: "For example, by lowering our P value threshold by only a factor of 10, the fraction of IGH genes with usage associated to at least one genetic variant increased from 73% to 91% (74/81)".

2.4. Similarly this paragraph ends with a description of the supplementary table (lines 226-231), how about putting that into a Supplementary Table legend and simply writing that comprehensive summary statistics for these association tests is provided.

We added a table (Table 2) with the number of significant genes and variants identified through the gene usage QTL analysis.

2.5. Could the Supplementary Table contain standard errors as well as effect size estimates please? (These can be inferred from the P-value but having them stated would be better.)

The residual standard errors from the linear regression models have been added to the supplementary table S4.

2.6. A suggestion would be: combine paragraphs on line 208 and 283 into a single paragraph that says briefly how association analysis & conditional analysis was conducted. Then summarise the main findings in subsequent paragraphs, possibly with specific subheadings e.g. for non-SV-driven associations, as needed.

Thank you for this suggestion. We attempted to rewrite this section as suggested, but felt in the end that the details of each analysis were out of context in the narrative. We worry that this would limit the comprehension of key results for some readers. If the reviewer or editor feels strongly about this rewrite, however, we can make an additional attempt.

2.7. I should add that my comments 2.1-2.6 are largely stylistic i.e. up to authors preference. In general the material in this section is interesting and does get across the key points and has defensible methods.

We appreciate this input. We want our narrative to be as clear as possible, and are grateful for the reviewer taking the time to articulate these points. We hope that overall our edits have improved clarity.

2.8 Another comment I had on this section was that the authors do not consider possible confounding. This could be important because the dataset is quite diverse e.g. individuals of varying ethnicities and data generated in different experiments on different platforms and so some examination of whether there is evidence that the associations correlate with any of the other experimental covariates seems needed. In saying this I note that this is an observational study; as such it can't decide what the underlying causal pathway of any association actually is, and there is a strong a priori case that genetic variation might be the cause of gene usage variation. Hence, my suggestion would be to include a covariate analysis as supplementary information and, for any associations where a covariate seems to 'explain' or partly explain the association this should be noted as a possible alternate pathway. (Specifically I am not suggesting to alter the currently presented association tests to include covariates, and I don't think this would make sense to do.)

Thank you for this suggestion. Our current models include age and AIRR-seq method performed, but not ethnicity. However, we are currently expanding our study to include a diverse range of individuals across ethnicities. In future work, we will perform population-specific analysis to account for potential population-driven effects. As a first study exploring genetic associations with antibody gene usage, given the known genetic differences between populations, we did not want to exclude any potential true signals by over correcting the models. We were reassured by the fact that many of the signals observed fit our expectations: , e.g., genes within structural variants (SVs) being associated with the SV or a single nucleotide variant in high linkage disequilibrium with the SV. That said, we agree that these are important issues to ultimately address when larger cohorts are available. To take this head on, we specifically note in the discussion that we believe future work should consider other potential variables , such as genetic ancestry, positive/negative selection, age, B cell subset, and tissue.

3.1. My other comment is that the Discussion (lines 433-572) is very long! (5 1/2 pages double-spaced.). I suggest the authors cut this down especially where this simply duplicates results. I would say the author's results are strong enough that they don't need to spend time re-justifying their efforts or repeating results. E.g. I'm not sure the paragraph on line 447 is needed, lines 456-468 seem to repeat results, so do lines 475-487, etc..

Thank you for this suggestion. We have shortened the Discussion by removing redundant and unnecessary text.

4.1 Re: figures. The authors have gone for a 'every figure is a massive multi-panel' figure type of approach. There is lots of information in figures but I personally think that removing some of this to supplementary figures, adding clarifying wording / labels and simplifying would help the presentation. Concrete examples below:

[No comment]

4.2. For example, in Figure 1, are panels b-d and f really needed? (These SVs are shown in panel a. and the only additional info is length, so they are repeating information. How about printing the lengths in panel a). Is the SV on panel f also in panel a? I couldn't spot it or these IGHD genes - if not why not? Adding more space between panels would also help.

Panels a-d and f did contain duplicated information. We moved panels b-d to the supplement, and included two new plots (panel b and c) with the SV allele size and allele frequency.

4.3. Many of the panels could be improved with better labelling (including a title). For example, the two CDFs in panel h are labelled 'individuals' on y axis - they mean proportion of individuals, presumably. (I had to work hard to understand what these panels meant. I think it is: 'for each number of deleted gene alleles up to 50, this is the proportion of individuals in that population with that number or fewer deleted gene alleles'?)

We have added more labels throughout the figures. For example for panel g we added the title "Common SNVs and Common SNVs in dbSNP".

4.4. In Fig 1 panel j it's not immediately obvious what 'SNVs' and 'Common SNVs' are and the bracket seems to say some of the common SNVs are rare (reading the legend I see this is about what dbSNP says are rare, but that's nonobvious in the figure); panel l is just showing three numbers (I think?) so doesn't really need to be there at all.

We changed figure 1 panel j to pie charts. We hope that these make the results more clear. Figure 1 panels f to h include the SNVs, indels and smaller SVs that are included in the QTL analysis. We have left these subpanels in the main figures as it helps the reader understand the variants used for the gene usage QTL analysis.

4.5. Fig 2 panel a confused me too. The first four tracks are visually similar and reflect somewhat related information so it's not immediately obvious what to look at first. Then, they use colours that mean different things in each track but have overlapping colour schemes. It's not stated what the P-value used is (is it the smallest P-value across variants for the gene). I assume the 'usage fold change' is an effect size derived from the regression estimate, but I was expecting to see this presented as a point and 95% confidence interval instead of a bar. The 'variant type' also doesn't need to be bars at all (the height is irrelevant) and the use of overlapping colours is again confusing.

We split panel a into three panels. We also added titles to each of the panels to improve the clarity of the panels. We also changed the color scheme so that the colors do not overlap within a subpanel. We hope that this addresses the reviewer's concerns.

4.6. Fig 2 panel e looks cool but (in common with several other panels) there's no indication of what it is about without reading the legend. (In this case, the legend for panel e refers to panel f as well which isn't there (at least in the combined pdf I'm using for this review). Similarly the Fig 3 panels don't give any indication what they relate to without reading the legend.

Figure 2 panel e were moved to a new figure (Figure 3), and titles were added to figures.

4.7. In short - I think the figures would benefit from an editorial review of the figures to substantially simplify and clarify. (Similar to many of my comments this is obviously stylistic rather than about the results themselves, which seem strong.)

We have improved the clarity of multiple figures by removing subpanels, adding titles, splitting some larger figures into multiple ones, and relocating some subpanels to the supplement. Hopefully, this has helped.

5.1 Line 758 "The datasets generated during and/or analysed during the current study are available from the corresponding author on reasonable request." -> please can the authors deposit the data, ideally including the haplotype-resolved assemblies and AIRR-seq results, in an open repository and include details here?

We have deposited all of the PacBio data in SRA, and the majority of AIRR-seq data is currently in SRA. We are waiting for collaborators to deposit the remaining set of AIRR-seq data.

Reviewer #2 (Remarks to the Author):

Comments on the manuscript “Genetic variation in the immunoglobulin heavy chain locus shapes the human antibody repertoire” by O.L. Rodriguez et al. (2022) submitted to Nature Communications

Rodriguez et al. in their manuscript “Genetic variation in the immunoglobulin heavy chain locus shapes the human antibody repertoire” show that a person’s antibody repertoire is strongly affected by genetic variation in the IGH locus. It is an impressive study, which combines germline IGH long-read sequencing data and repertoire (AIRR-seq) data from the same individuals. The studied cohort includes individuals of various ethnicities, which gives this study another dimension. The results presented in this study are of great importance to the field and provide new insights into how antibody repertoires are generated.

I do not have experience with long-read sequencing, only with AIRR-seq; however, the methods and analytical approaches sound reasonable to me.

The conclusions of the study are supported by the data presented.

Although the code for the entire analysis is not made publicly available (apart from one custom python script), the authors provide some details about the packages and functions used, as well as modifications to the default parameters. Perhaps a bit more detail could be included in the methods sections “Gene usage QTL analysis”; “Network analysis of variants associated with multiple genes” and “Regulatory analysis”.

Some minor edits might improve clarity and readability:

Samples used in the study: In the Materials and Methods section, the authors say that PBMCs or PMN were obtained from either Stanford University, Harvard University or STEMCELL Technologies. Could you add the information about the cell source and the institution from which the samples were obtained to the Supplementary Table 1?

The cell source and institution were added to Supplementary Table 1. See columns: “Source” and “Cell source for PacBio”.

Clone definition: How were the clones defined in the AIRR-seq analysis? Can the authors make it more clear in the results section how they defined a clone? The criteria for defining a clone can vary between different studies.

We added this statement to the results section: “Similar sequences with the exact junction length, V and J allele were grouped into clones (Methods)”. In the methods section we explicitly state how we assigned clones. We included the tools and the parameters used. Briefly, the tools and parameters used were the recommended parameters from Immcantation.

“Clones were detected using the modified Change-Os with the `shazam distToNearest` command and `model='ham', normalize='len` parameters, `shazam findThreshold (parameters: method='gmm`,

model='gamma-gamma`), and `DefineClones.py (parameters: -act set -model ham -norm len -mode allele)` commands.”

Figures: The figures are sometimes too complex and/or difficult to read. Please, reconsider if all parts of all figures are absolutely necessary, or perhaps if it is possible to show certain data differently:

- Figure 1; mainly a) and h): The color scheme the authors use in Fig. 1a) and h) makes it very difficult to distinguish the data shown. Different shades of orange are difficult to distinguish both on screen and when printed; the same goes for the different shades of teal.

We have made several improvements to Figure 1 to increase clarity. We simplified the figure by removing some of the unnecessary annotations and subpanels (b-f), which were moved to the supplement. Additionally, we ensured that the same colors were not used throughout the figures to prevent confusion. We believe that these changes have improved the overall clarity of the figure.

- Figure 1 j) and l): In Fig. 1j) the authors use the color red for “No”, but in Fig. 1l) they use the same color for “Yes”. For the sake of consistency and to make it easier for the readers, it might be good to keep the same colors for “Yes” and “No” in the same figure. However, I am not sure if these bar plots are absolutely necessary to be presented in this way. Would it be perhaps better to have these as tables rather than bar plots, so that it is easier to see the exact values?

We have removed the color Red from figure 1j. We agree that colors should be consistent. We tried to ensure that the same colors are not used multiple times.

- Figure 2 – part f) is missing: I am not sure if it is due to the file merging, but I cannot see part f in Figure 2, despite it being mentioned in the figure legend. I can only see a)-e).

We apologize for this. We aren't sure what happened to figure 2f. We have modified figure 2 as part of our revision.. We added titles and removed some of the box plots. We hope this makes the figures easier to understand.

- Figure 3d): This plot is not easy to read. In the results section (line 320) it says: “We found 2607 (66%) guQTL variants associated with > 1 gene (Fig. 3d)”; however, this information is not easy to see from figure 3d. Could the actual values be perhaps put on top of the bars in the barplot? The x-axis labels are also a little bit confusing. Having the x-axis labels as 1, 2, 3, etc. would be easier to read than 2.5, 5.0, 7.5, etc. More labels on the y-axis might also be helpful.

We added a title and expanded on the x- and y-axis labels. Thank you for this suggestion. We believe this makes the figure clearer.

- Figure 3e) and f): Would it be possible to make these plots bigger and put them into a separate figure? With their current size, the text is too small to read. Also, with the overlapping text, the genes names (especially those in the middle) are hard to see.

We changed the color of the node to make the node label more legible. We've also increased the node text size, made it bold and reduced the edge text size.

Discussion

Line 538: In what way does the reference no. 91 (Ohlin (2020) Front. Immunol.) support your statement? I had a look at the referenced paper, but I am unsure how it relates to the point being discussed in lines 536-539.

Ohlin 2020 showed that specific alleles of genes expressed at a very low frequency contained a different set of amino acids (therefore also genetic variants) compared to other alleles within the gene subfamily. This demonstrates a connection between coding variation and allele usage, albeit, this was just observational in the Ohlin et al. study due to the limited number of samples. In our manuscript, we expanded on this observation, as our sample size allowed for us to statistically test for associations between allele usage and coding region allelic variants (or amino acid composition).

To make this point clearer we have edited this section to read:

“Our analysis revealed that IGHV coding variation was in many cases linked to guQTLs, supporting previous reports indicating that usage patterns can coincide with amino acid differences^{23,41,91}, including those that are important for Ab-antigen interactions in infectious disease responses^{23,41}.”

Data Availability: What is the reason for not making the data publicly available in a repository?

We have deposited all of the PacBio data in SRA, and the majority of AIRR-seq data is currently in SRA. We are waiting for collaborators to deposit the remaining set of AIRR-seq data.

Supplementary data:

- Would it be possible to label the figures or at least number the pages, so that it is easier to distinguish which figure is which?

We added the figure legends under the figures.

- Could the figure legends appear under the figures, at least for those figures that do not take up the entire page?

We added the figure legends under the figures.

- Figure 11: the figure itself is not very clear, and the figure legend is not informative either.

Figure 11 is now supplementary figure 12. More text was added to the figure legend. Hopefully the figure is more clear.

- Figure 13: Same as in Fig.3 in the main text; the gene names and the numbers are very difficult to see. Especially the text that lies directly on top of those dots is almost impossible to see.

The gene names were increase in Figure 13 (now figure 14). We also increased the resolution so that the gene names are more clear if the reader zooms in.

Apologies for the delay in reviewing this manuscript and wishing the authors happy holidays!

Reviewer #3 (Remarks to the Author):

the study reported here is of immense importance to immunology and the study of multivariate gene areas. The authors have applied cutting edge sequencing methods and sequence analysis methods to characterize the heavy chain germline locus in its entirety in 154 individuals. The novel data set by itself should have a great impact on the field as it is nearly unique in describing this locus. This dataset radically increases our knowledge of the true makeup of germline diversity. Over and above these findings the authors also show (through sequencing and analyzing of somatic repertoires from these individuals) that there is a relationship between the genotype (And number of copies in the germline) of a different gene types and their somatic expression. Moreover, we can see from this study how frequent some but not all SV and SNV are we can now start to characterize what part of diversity in germline populations influences somatic germline gene usage and expression.

There are no major issues with the paper but i have a few minor ones that should be addressed before publication:

1) A glossary of acronyms should be made of all acronyms from SV and SNV to QTL and its types.

Thank you for the suggestion, we will gladly add this. We will ask the editor on how to best approach this.

2) the use of p-values to describe effect size. This is a bad and misleading practice that should not remain in the paper.

thus for instance in figure 5 genes should be ranked by the skew in their expression not by their p value. The description in the figure caption also is misleading. There is no such thing as "3 most significant". As the authors themselves state all genes colored in pink are significant with significance set by the p value threshold - $p < 0.05$. thus they are all "equally" significant. The exact p values could be influenced by effect size in terms of size of skew but also by other things.

other figures suffering from this issue are

figure 2a and figure 3, b,c and g and supplemental figure 7

We apologize for any confusion here. We use the beta value of the linear regression to describe the effect size. The beta corresponds to the slope of the linear model. We use this to represent the effect of adding a single alternate SNV, SV, or indel allele on the usage.

We use a p-value to simply rank our associations because the p-values represent the probability that a statistical summary of our data under the linear regression would be equal to or more extreme than the observed data. But this by no means means that we are ranking the "biological importance" of these associations, only that we are using statistical significance as an approach to select and display specific associations. While we recognize that there are limitations to using p-values to "biologically rank", we would argue that ranking SNV associations by their p-value is standard practice in genetic association analysis and has been demonstrated to identify causative associations. Given the sample size in our study, we understand that (and are forthright about this in the discussion) that follow-up analyses in larger cohorts, with experimental validations will be required to refine these associations. This point in particular is additionally motivated by our inclusion and discussion conditional effects and gene-variant networks, as we believe that the genetic determinants of usage for many genes will be complex.

3) the network figures and especially in the sup. materials are indecipherable. it is very hard to read the gene names and numbers describing the strengths of edge connection. could the author redraw the networks and especially figure 12 where one can not read the gene names.... they do not need to make this figure a printable page size. make it as large as needed so we can see all the gene names.

We have re-made the network figures to make them more legible. We changed the color of the node to make the node label more legible. We've also increased the node text size, made it bold and reduced the edge text size. We also split the cliques into multiple pages. This has made the networks more decipherable.

4) the figure captions in sup materials are lacking. please describe these figures with as much detail as the figures in the paper.

Figure legends were expanded for the supplemental figures.

REVIEWER COMMENTS

Reviewer #1 (Remarks to the Author):

Thanks to the authors who have addressed the majority of my points.

They have not addressed the point about potential confounding by ethnicity, but have given strong reasons for that which I'm satisfied with.

I still have two comments which are again about presentation:

1 I feel duty bound to say that in the presentation there is still a very prominent focus on 'significance' of associations - in the text, tables, and figures. This point relates to my earlier point 2.2, to wit "the chosen P-value threshold (based on Bonferroni) is a pragmatic rather than especially meaningful threshold", and to the comment made by another reviewer. Although using significance this way is common, it has multiple issues:

i. statistical 'significance' isn't a biologically or scientifically meaningful quantity in itself, it's an ad hoc but practically useful choice of threshold. So it should be used that way not presented as of primary interest.

ii. the word 'significant' is unhelpfully overloaded (e.g. scientific vs. statistical significance), and to me this adds confusion. In many cases simply saying $P < \text{threshold}$ would be clearer. For example, the abstract and parts of the introduction use 'significantly' in what I think is the broader scientific sense.

iii. quantities like the effect size and variant frequency are really much more relevant to understanding an association, once an association has compelling evidence.

iv. Formally speaking, I don't *think* there's such a thing as 'more significant' - as the other reviewer pointed out. That is, in the formalism either the P-value is below the preselected threshold or not. (This is another argument for not relying on a the 'statistical significance' formalism.)

The reason I'm raising this again is that I personally still find this emphasis to be very prominent in the current manuscript, and think it would be improved by reviewing this language / emphasis - particularly as this is likely to be an influential paper. Examples:

- Table 2 is called 'Number of significant variants'. (It means the number with $P < 9E-6$, at very least it should say 'Number of variants meeting statistical significance')
- Figure 2 presents result with a separate colour for variants with $P < \text{threshold}$ - even though the y axis is already the $-\log_{10}$ P-value. Surely this over-emphasises the threshold? A less prominent dashed horizontal line would be better. (NB. the 'usage fold change' part of this figure also uses both y axis and a categorised colour scale for the same thing, which feels odd to me.)
- Fig 3c ditto, I don't believe the red/orange add anything useful but it looks quite prominent, a horizontal line would be better but frankly I don't think the P-value threshold is particularly meaningful or needed on this plot.
- Fig 4c ditto
- Fig 7a has a colour for 'significant - No/Yes', which again doesn't add anything.
- Lastly there are multiple usages of 'significant' in the text, either called 'statistical' or not, many of which could be better worded by referring to the strength of evidence or effect size.

As an example, this: 'These SV associations were among the most statistically significant in this dataset' could be altered to something like:

'These were among the strongest associations observed in this dataset ($P < XX$) ... The association with the lowest P-value was for IGHV3-64D (P-value = XX , at frequency X ; effect size (95% CI) = XX (yy-zz)), ...'

which I think is more informative and gets rid of the awkward 'most significant' part. Similarly elsewhere.

For clarity, this comment is minor because it is mostly about presentation of results, not about the statistical analysis itself which seems well conducted, I'd leave it up to the authors & editor to consider this further.

2. Some of the figure presentation is still a bit problematic, e.g.:

- I don't see why bar plots for $-\log_{10}$ P-value, R^2 and usage fold change in Figure 2 make sense. Bar plots make most sense for count type data. Why not plot as points - for the usage change this could also show confidence intervals.

- Similarly Fig 7a, bar plots for P-values seems odd to me.

- Fig 2c is a bit odd because it is truncated at the top of the plot. Why not show log₁₀ fold change instead and avoid the thresholding?

- As in my comment above - using both colour and y axis scale to denote the same thing seems a bit odd to me. If you want to distinguish categories, use horizontal lines I think?

- Several of the colour schemes are bright and/or use similar adjacent colours. Fig 2c uses two similar shades of blue for example; Fig 7 similarly has hard-to-distinguish colours (that look a lot like the default ggplot colour scheme). Other figures also have very bright colour palettes. I like how Fig 5 colours work but it also has nearby blues. There are lots of colour palettes available e.g. the Brewer palette or Wes Anderson palette sets that are useful as a starting point for these types of plot, I suggest tuning the palette for each plot to draw out the main distinctions.

- The new plots have a *lot* of very similar-looking box plots. These are important but I think most of them (say Fig 5 onwards) could reasonably be included as supplementary plots.

- Fig 8 has a table in a figure. And one panel is a plot from UCSC genome browser, not about their data. I'd personally suggest moving the box plots to supplementary and just including the table.

These comments notwithstanding, thanks for the opportunity to review this paper!

Reviewer #2 (Remarks to the Author):

Review of revised manuscript NCOMMS-22-35083A:

I would like to thank the authors for addressing my comments. The clarity of the figures has significantly improved. The supplementary figures are now much easier to read and understand with the figure legends directly below them.

There are still one major and a couple of minor issues I missed earlier:

- In Supplementary Figure 10., the x-axis labels 0 and 1 (Genotypes) have probably been switched (the plots show opposite of what is described in the text):

- The data does not support the legend and the statements made in the results section. The legend says the gene usage is lower in individuals with the deletion, but the plots show the opposite. The ones with genotype 0 (which according to the legend represents the deletion) show higher gene usage, which seems unlikely considering 'deletion' should mean no expression of that gene.

- Does "deletion" here refer to a deletion on at least one chromosome or both?

- Also, how many individuals were included in these box plots?

- Do the horizontal lines in all the boxplots in this article represent the mean or the median value?

- Code availability statement is missing from the main article file. In the Editorial Policy checklist, the authors have checked that they have provided code availability statement in the manuscript, but I cannot see one in the file that was provided to me. In one of the Methods subsections, there is a link to a custom script, but I could not find the links for scripts for the other parts of the analysis.

There are also two more things I was wondering about, but these are very minor and will not significantly affect the quality of the manuscript, in my opinion. The authors can choose to ignore these two comments below:

- Supplementary Figure 11. The x-axis of the plot shows the number of copies. Why are the values '1'; '2'; '3'; '4'; and '4 or 5'? Why is there '4 or 5'? Is it so that the copy number cannot be reliably determined in the case of more than 4 copies? I might have missed it, but I could not find further explanation in the text.

- Supplementary Figure 12. It is slightly confusing that in this figure, genotype 2 refers to a homozygous deletion and genotype 0 means reference, but in Suppl.Fig. 10 'genotype 0' refers to a deletion. It would be more helpful to refer to the genotypes similarly to how it is done in Fig. 3, where a homozygous deletion is referred to as DEL/DEL and the homozygous reference is referred to as REF/REF.

Reviewer #3 (Remarks to the Author):

All my comments have been addressed i recommend accepting this paper.

Response to Reviewers

Reviewer #1 (Remarks to the Author):

Comment 1: I feel duty bound to say that in the presentation there is still a very prominent focus on 'significance' of associations - in the text, tables, and figures. This point relates to my earlier point 2.2, to wit "the chosen P-value threshold (based on Bonferroni) is a pragmatic rather than especially meaningful threshold", and to the comment made by another reviewer. Although using significance this way is common, it has multiple issues:

- i. statistical 'significance' isn't a biologically or scientifically meaningful quantity in itself, it's an ad hoc but practically useful choice of threshold. So it should be used that way not presented as of primary interest.
- ii. the word 'significant' is unhelpfully overloaded (e.g. scientific vs. statistical significance), and to me this adds confusion. In many cases simply saying $P < \text{threshold}$ would be clearer. For example, the abstract and parts of the introduction use 'significantly' in what I think is the broader scientific sense.
- iii. quantities like the effect size and variant frequency are really much more relevant to understanding an association, once an association has compelling evidence.
- iv. Formally speaking, I don't *think* there's such a thing as 'more significant' - as the other reviewer pointed out. That is, in the formalism either the P-value is below the preselected threshold or not. (This is another argument for not relying on a the 'statistical significance' formalism.)

The reason I'm raising this again is that I personally still find this emphasis to be very prominent in the current manuscript, and think it would be improved by reviewing this language / emphasis - particularly as this is likely to be an influential paper. Examples:

>>> We appreciate the reviewer raising this point of concern, and we regret that our wording has given the reviewer the impression that we are conflating statistical significance with biological significance. We have refined our wording throughout, and clarified the fact that we use statistical significance as means to partition our data for the purposes of presentation (as is commonplace in the literature). Our reasoning for providing comprehensive summary figures (e.g., Figure 2) and tables (Supplemental Tables) is so that the reader can take the data for what they are, without us telling them what exactly is biologically significant. We are simply using the P value thresholds to state the claim that within this specific dataset, these genes and variants have statistical support, even after multiple-testing correction. Our goal is to use these data to illuminate an important phenomenon that has been largely ignored by the immunology community. We know that follow up work will be required to exhaustively understand the impacts of IGH variants on the repertoire, pin down the “biology”. And we fully expect that as future work is conducted, that many of the genes not currently meeting the stringent P value thresholds used here will in fact be brought into the picture. In summary, we hope that our edits sufficiently address your concerns, and that they will make our viewpoints and interpretations of the data clear to the readership of Nature Communications.

In addition, we have tried to address several of the specific points below.

- Table 2 is called 'Number of significant variants'. (It means the number with $P < 9E-6$, at very least it should say 'Number of variants meeting statistical significance')

>>> Thanks for pointing this out. We see how this word choice could be misconstrued. We have made this suggested edit.

- Figure 2 presents result with a separate colour for variants with $P < \text{threshold}$ - even though the y axis is already the $-\log_{10}$ P-value. Surely this over-emphasises the threshold? A less prominent dashed horizontal line would be better. (NB. the 'usage fold change' part of this figure also uses both y axis and a categorised colour scale for the same thing, which feels odd to me.)

>>> We had initially used a dotted line in this figure, but ultimately opted for distinct colors here so the distinction was clear. Again, we appreciate the reviewer's opinion here regarding the over-emphasis on the threshold. However, this plot is meant to give a high-level summary of the results, and the colors are making the distinction in genes passing the threshold or not, so that they are consistent with the numbers in the text. Any reader will have the option to assess the data directly from the supplemental tables, if they wish to get more detail on the statistics for any given gene.

- Fig 3c ditto, I don't believe the red/orange add anything useful but it looks quite prominent, a horizontal line would be better but frankly I don't think the P-value threshold is particularly meaningful or needed on this plot.

>>> As per the reviewer's suggestion, we have removed the color from this plot.

- Fig 4c ditto

>>> Colors have been removed.

- Fig 7a has a colour for 'significant - No/Yes', which again doesn't add anything.

>>> Colors have been removed.

- Lastly there are multiple usages of 'significant' in the text, either called 'statistical' or not, many of which could be better worded by referring to the strength of evidence or effect size.

As an example, this: 'These SV associations were among the most statistically significant in this dataset' could be altered to something like:

'These were among the strongest associations observed in this dataset ($P < XX$) ... The

association with the lowest P-value was for IGHV3-64D (P-value = XX, at frequency X; effect size (95% CI) = XX (yy-zz)), ...'

which I think is more informative and gets rid of the awkward 'most significant' part. Similarly elsewhere.

>>> Thank you for these suggestions. As per our response above, we have edited these sentences to reflect this point.

For clarity, this comment is minor because it is mostly about presentation of results, not about the statistical analysis itself which seems well conducted, I'd leave it up to the authors & editor to consider this further.

2. Some of the figure presentation is still a bit problematic, e.g.:

- I don't see why bar plots for $-\log_{10}$ P-value, R2 and usage fold change in Figure 2 make sense. Bar plots make most sense for count type data. Why not plot as points - for the usage change this could also show confidence intervals.

>>> We appreciate the reviewers comment here, but not sure it is critical. We opted for bars, as we are (1) plotting discrete values (e.g., we are just showing p values and R2 values between genes), and (2) were concerned that a point representing these values for every gene would be harder for the reader to visualize in the figure.

- Similarly Fig 7a, bar plots for P-values seems odd to me.

>>> We removed this panel.

- Fig 2c is a bit odd because it is truncated at the top of the plot. Why not show \log_{10} fold change instead and avoid the thresholding?

>>> The plot is truncated because some values are infinity, resulting from the fact that a few of the genes residing in deletions have 0% usage in the repertoire in individual homozygous for the deletion. Using \log_{10} would still produce infinite values in these cases.

- As in my comment above - using both colour and y axis scale to denote the same thing seems a bit odd to me. If you want to distinguish categories, use horizontal lines I think?

- Several of the colour schemes are bright and/or use similar adjacent colours. Fig 2c uses two similar shades of blue for example; Fig 7 similarly has hard-to-distinguish colours (that look a lot like the default ggplot colour scheme). Other figures also have very bright colour palettes. I like how Fig 5 colours work but it also has nearby blues. There are lots of colour palettes available

e.g. the Brewer palette or Wes Anderson palette sets that are useful as a starting point for these types of plot, I suggest tuning the palette for each plot to draw out the main distinctions.

>>> Thanks for pointing out this issue. We have made edits to the colors in these figures to address this comment.

- The new plots have a *lot* of very similar-looking box plots. These are important but I think most of them (say Fig 5 onwards) could reasonably be included as supplementary plots.

- Fig 8 has a table in a figure. And one panel is a plot from UCSC genome browser, not about their data. I'd personally suggest moving the box plots to supplementary and just including the table.

These comments notwithstanding, thanks for the opportunity to review this paper!

Reviewer #2 (Remarks to the Author):

Review of revised manuscript NCOMMS-22-35083A:

I would like to thank the authors for addressing my comments. The clarity of the figures has significantly improved. The supplementary figures are now much easier to read and understand with the figure legends directly below them.

>>>We thank the reviewer for their continued positive feedback.

There are still one major and a couple of minor issues I missed earlier:

- In Supplementary Figure 10., the x-axis labels 0 and 1 (Genotypes) have probably been switched (the plots show opposite of what is described in the text):
 - The data does not support the legend and the statements made in the results section. The legend says the gene usage is lower in individuals with the deletion, but the plots show the opposite. The ones with genotype 0 (which according to the legend represents the deletion) show higher gene usage, which seems unlikely considering 'deletion' should mean no expression of that gene.

>>> Thank you for pointing out this error. We have edited this figure legend to address all of these comments/questions.

- Does "deletion" here refer to a deletion on at least one chromosome or both?

>>> One chromosome. The genotype 1 (now REF/DEL) corresponds to a deletion on a single haplotype.

- Also, how many individuals were included in these box plots?

>>> 7 individuals are included in the REF/DEL box plot, and all other samples are in the REF/REF boxplot. We added “n=7” to the figure legend.

• Do the horizontal lines in all the boxplots in this article represent the mean or the median value?

>>> Apologies for neglecting to include this information. We have now indicated this in each figure legend.

C

• Code availability statement is missing from the main article file. In the Editorial Policy checklist, the authors have checked that they have provided code availability statement in the manuscript, but I cannot see one in the file that was provided to me. In one of the Methods subsections, there is a link to a custom script, but I could not find the links for scripts for the other parts of the analysis.

>>> We have now included this information in the manuscript and provided appropriate links to our code.

There are also two more things I was wondering about, but these are very minor and will not significantly affect the quality of the manuscript, in my opinion. The authors can choose to ignore these two comments below:

• Supplementary Figure 11. The x-axis of the plot shows the number of copies. Why are the values ‘1’; ‘2’; ‘3’; ‘4’; and ‘4 or 5’? Why is there ‘4 or 5’? Is it so that the copy number cannot be reliably determined in the case of more than 4 copies? I might have missed it, but I could not find further explanation in the text.

>>> Yes, the reviewer is correct. We have added this to the figure legend.

• Supplementary Figure 12. It is slightly confusing that in this figure, genotype 2 refers to a homozygous deletion and genotype 0 means reference, but in Suppl.Fig. 10 ‘genotype 0’ refers to a deletion. It would be more helpful to refer to the genotypes similarly to how it is done in Fig. 3, where a homozygous deletion is referred to as DEL/DEL and the homozygous reference is referred to as REF/REF.

>>> We appreciate the reviewer pointing this out, and we agree it is confusing. We have edited our figures to indicate what the SV genotypes are, rather than referring to them as “0”, “1”, and “2”.

Reviewer #3 (Remarks to the Author):

All my comments have been addressed i recommend accepting this paper.

REVIEWERS' COMMENTS

Reviewer #2 (Remarks to the Author):

All of my comments have been sufficiently addressed, and I recommend this manuscript to be accepted for publication. Thank you for the opportunity to review this article!